# Improving mesoscale wind speed forecasts using LiDAR-based observation nudging for airborne wind energy systems

Markus Sommerfeld[1], Martin Dörenkämper[2], Gerald Steinfeld[3], and Curran Crawford[1]

[1]Institute for Integrated Energy Systems, University of Victoria,British Columbia, Canada
[3]Institute of Physics-Energy Meteorology, Carl von Ossietzky Universität Oldenburg, Germany
[2]Fraunhofer Institute for Wind Energy Systems, Oldenburg, Germany

*Correspondence to:* Markus Sommerfeld (msommerf@uvic.ca)

**Abstract.**

Airborne wind energy system (AWES) aim to operate at altitudes above conventional wind turbines where reliable high-resolution wind data is scarce. Wind light detection and ranging (LiDAR) measurements and mesoscale models both have their advantages and disadvantages when assessing the wind resource at such heights. This study investigates whether assimilating measurements into the mesoscale weather research and forecasting (WRF) model using observation nudging generates a more accurate, complete data set. The impact of continuous observation nudging at multiple altitudes on simulated wind conditions is compared to an unnudged reference run and to the LiDAR measurements themselves. We compare the impact on wind speed and direction for individual days, average diurnal variability and long-term statistics. Finally, wind speed data is used to estimate optimal traction power and operating altitudes of AWES. Observation nudging improves the WRF accuracy at the measurement location. Close to the surface the impact of nudging is limited as effects of the air-surface interaction dominate, but becomes more prominent at mid-altitudes and decreases towards high-altitudes. The wind speed frequency distribution shows a multi-modality caused by changing atmospheric stability conditions. Therefore, wind speed profiles are categorized into various stability conditions. Based on a simplified AWES model the most probable optimal altitude is between 200 and 600 m. This wide range of heights emphasizes the benefit of such systems to dynamically adjust their operating altitude.

Keywords: Airborne Wind Energy, Wind Measurement, Onshore Wind, Weather Research and Forecasting Model, Observation Nudging, Statistic Wind Conditions, LiDAR, WRF

# 1 Introduction

The prospects of higher energy potential and more consistent strong winds and less turbulence in comparison to near surface winds sparked the interest in mid-altitude, here defined as heights above 100 m and below 1500 m, wind energy systems. airborne wind energy system (AWES) are a novel class of renewable energy technology that harvest stronger winds at altitudes which are unreachable by current wind turbines, at potentially much reduced capital cost (Lunney et al., 2017; Fagiano and Milanese, 2012). For practical and economical reasons we focus on resource assessment within the lower part of the atmosphere, an altitude range spanned by the highly-variable boundary layer. Unlike conventional wind energy which has converged to a single concept with three blades and a conical tower, several different AWES designs are under investigation by numerous companies and research institutes worldwide (Cherubini et al., 2015). Various concepts from ring shaped aerostats, to rigid wings and soft kites with different sizes, rated power and altitude ranges compete for entry into the market. Since this technology is still in an early stage, none are currently commercially available.

Developers and operators of large conventional wind turbines, AWES and drones require accurate wind data to estimate power output and mechanical loads. They currently rely on oversimplified approximations such as the logarithmic wind profile (Optis et al., 2016) or coarsely resolved reanalysis data sets (Archer and Caldeira, 2009; Bechtle et al., 2019) as the applicability of conventional spectral wind models (Burton, 2011) have not been verified for these altitudes. First investigations (Fechner and Schmehl, 2018) resorting to the Mann model (Mann, 1994; IEC, 2005) have been conducted.

Recent advancements in wind LiDAR technology enable measurements at higher altitudes. This measurement technique however suffers from reduced data availability with increasing altitude caused by a decrease in aerosol density which is needed for the backscattering of the LiDAR signal (Peña et al., 2015). No mid-altitude measurement device can reliably gather long-term, high-frequency data. Temporal and spatial resolution of LiDAR devices is insufficient to precisely measure high-frequency fluctuations, but estimated turbulence intensity correlates with sonic turbulence measurements for lower altitudes (Sathe et al., 2011). Balloon mounted sonic anemometer are in early development (Canut et al., 2016). The expensive and time consuming nature of measurements motivates the usage of numerical weather prediction (NWP) models such as the mesoscale model weather research and forecasting (WRF) as an adequate tool to assess synoptic characteristics of the atmospheric boundary layer (ABL) (Al-Yahyai et al., 2010). These models typically have a spatial resolution that ranges from one kilometer to tens of kilometers and a temporal resolution in the order of minutes. Sub-gridscale high-frequency variations of resolved quantities are parameterized. Mesoscale models can be used to produced long-term reference data sets up to higher altitudes such as the New European Wind Atlas (Witha et al., 2019).

This work is a continuation of a previous investigation of mid-altitude wind LiDAR measurements (Sommerfeld et al., 2019). The measurements used in these studies were gathered as part of the *OnKites II* project (Gambier et al., 2017) at the Fraunhofer institute for wind energy systems (IWES) with the goal of evaluating the potential of AWES. This paper makes use of various statistical tools to describe the relationship between the mesoscale WRF model and LiDAR measurements to determine the impact of wind speed observation nudging (Mylonas-Dirdiris et al., 2016).

Section 2 describes the measurement campaign. Section 3 introduces the mesoscale model and observation nudging methodology used in this article. Section 4 quantifies the impact of observation nudging and summarizes the statistical differences between WRF and LiDAR. Results are applied to estimate optimal operating altitude and power output based on a simplified AWES model in section 4.7. Section 5 concludes the article with an outlook and motivation for future work.

## 2   Measurement Campaign

The LiDAR data used in this study (Bastigkeit et al., 2017) were collected between September 1st, 2015 and February 29th, 2016 at the 'Pritzwalk Sommersberg' airport (Coordinates: Lat: 53° 10' 47.00"N, Lon: 12° 11' 20.98"E) in Northern Germany (see white X in figure 1). The area surrounding the airport mostly consists of flat agricultural land with the town of Pritzwalk to the South. A *Galion4000* single beam pulsed wind LiDAR from SgurrEnergy was used (Gottschall et al., 2009). Wind speed data were collected using the doppler beam swinging (DBS) method (opening angle of 62°) which averaged multiple line of sight measurements at constant elevation angle and four azimuth angles to calculate the 10 min mean wind speed at 40 range gates up to an altitude of about 1100 m. Reference measurement found the mean LiDAR error to be around 1% with a standard deviation of 5% (Gottschall, 2013). The resulting wind speed is inherently spatially and temporally averaged. At an altitude of 1100 m the radius of the averaging disc defined by the four azimuth positions with 90° increments is about 585 m. For the reconstruction of 10 min mean wind speed it is thus assumed that the wind vector does not change over this area, a valid assumption for these heights over flat terrain.

LiDAR data availability highly depends on the applied carrier-to-noise ratio (CNR) filter and the aerosol content of the air as the wind speed is calculated based on the backscatter of the emitted laser beam. Most aerosols originate from the surface and are transported aloft. Particle density decreases with height and drops to almost zero within the free atmosphere above the ABL (Matthias and Bösenberg, 2002). Data quality quantified by the CNR dropped on average by approximately 5 dB over the course of 1000 m. A fixed CNR threshold of $CNR_{dB} > -25$ dB combined with additional self-defined filters (Sommerfeld et al., 2019) were applied and insufficient data was discarded. As a result, data availability dropped from about 81% at 100 m and about 24% at 1000 m. Low data availability caused by weather effects (e.g. strong precipitation) further emphasizes the importance of simulations for mid-altitude wind resource assessment as no measurement technique with sufficient spatial and temporal resolution is available at this point.

## 3 Mesoscale Modeling Framework

To complement the 6 months LiDAR data set two WRF 3.6.1 simulations using the advanced research weather research and forecasting (ARW) model (Skamarock and Klemp, 2008) were carried out. The 'baseline run' , which is hereinafter referred to as *NoOBS*, is a 12 month study of the area around the measurement location (see figure 1) from the 1st of September 2015 used to derive annual statistics. LiDAR measurements (Sommerfeld et al., 2019) were incorporated into the six months test model between September 2015 and February 2016 using *OBSGRID* (Wang et al., 2015), which is hereinafter referred to as *OBS*.

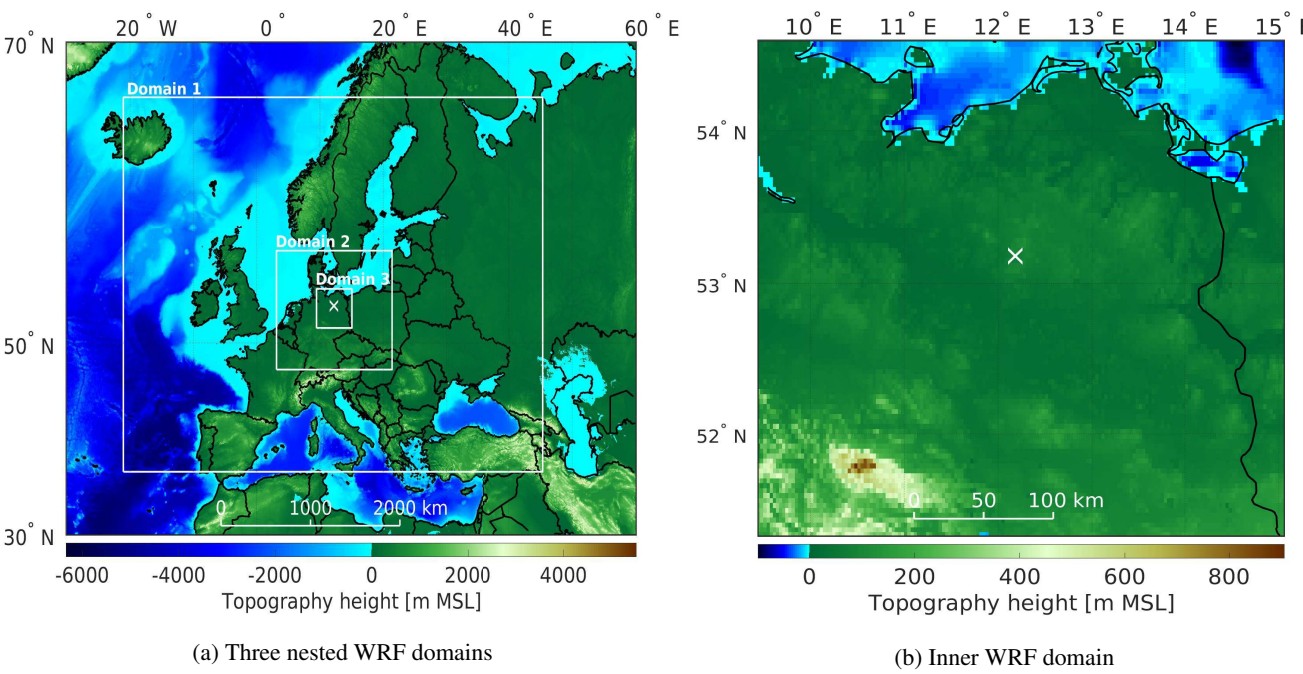

(a) Three nested WRF domains

(b) Inner WRF domain

**Figure 1.** Topography map of all three WRF model domains (a) and a magnification of the innermost domain (b) with the LiDAR measurement site marked by a white X.

This methodology uses the difference between model and measurements to calculate a non-physical forcing term which is added to the governing conservation equations of the simulation to gradually nudge the model towards the observation (see equation 1) (Stauffer et al., 1991; Deng et al., 2007). Each simulation is composed of three nested domains with 27-, 9- and 3-km grid spacing and horizontal grid dimensions of about $120 \times 120$ elements at 60 heights along the terrain following vertical hybrid pressure coordinate $\eta$. Differences between the simulation runs (see section 3.1) are compared within the innermost domain of the simulation. Output data was stored in 10 min intervals. Figure 1 shows the topography map of the simulation. Initial and boundary conditions of both simulations are based on the *ERA-Interim* (Dee et al., 2011) reanalysis data set by the European centre for medium-range weather forecasts (ECMWF) which consists of 6 hourly atmospheric fields with a spatial resolution of roughly 80 km horizontally and 60 $\eta$ levels. Turbulent Kinetic Energy closure within the ABL was achieved by using the Mellor Yamada Nakanishi Niino (MYNN) 2.5 scheme which predicts sub-grid turbulent kinetic energy (TKE)

as a prognostic variable (Nakanishi and Niino, 2004; Lee and Lundquist, 2017). The Noah-MP land-surface model, MYNN surface layer scheme were used. The rrtm longwave radiation and Dudhia shortwave radiation scheme were used (see: table A1 in the appendix). In addition to observation nudging (see subsection 3.1) analysis nudging was performed on every domain of each simulation. Analysis nudging nudges each grid point towards a time-interpolated value from gridded analyses of
synoptic observations (Stauffer et al., 1991) whereas observation nudging directly drives the simulation towards the additional observations. Within the planetary boundary layer (PBL) of the inner domain analysis nudging was switched off (see nudging settings in table A1 in the appendix). All simulations were run on the *EDDY* [1] High-Performance Computing clusters at the University of Oldenburg.

## 3.1 Observation Nudging

Observation nudging also referred to as 'dynamic analysis' is a form of four-dimensional data assimilation (FDDA) where each grid point within the radius of influence and time window is nudged towards observations using a weighted average of differences between model ($q_m$ interpolated at the observation location) and observations ($q_o$) (Dudhia, 2012; Reen, 2016). In this study horizontal wind speed $U$ and direction $\Phi$ were nudged towards measurements with a time interval of six hours between an altitude of 66 m and 1100 m, in order to not overly constrain the simulation. Nudging could not be performed at
15  times and altitudes where LiDAR data was not available. The non-physical forcing term is implemented in form of prognostic equations (Deng et al., 2007):

$$\frac{\partial q\mu}{\partial t}(x,y,z,t) = F_q(x,y,z,t) + \mu G_q \frac{\sum_{i=1}^{N} W_q^2(i,x,y,z,t)\left[q_o(i) - q_m(x_i,y_i,z_i,t)\right]}{\sum_{i=1}^{N} W_q(i,x,y,z,t)} \tag{1}$$

$q$ refers to the quantity that is nudged, $\mu$ is the dry hydrostatic pressure, $F_q$(x,y,z,t) is the physical tendency term of $q$, $G_q$ is the nudging strength of $q$, $N$ is the total number of assimilated observations, $i$ is the index of the current observation, $W_q$ is the
weighting function based temporal and spatial separation between grid cell and observation (Dudhia, 2012). Four weighting functions $G_q$, $W_t(x,y,z,t)$, $W_z(x,y,z,t)$ and $W_{xy}(x,y,z,t)$ describe the temporal and spatial nudging strength. Values used in this study can be found in the appendix (table A1). The inverse of $G_q$ (here about $1/6 \ 10^{-4}s \approx 46$ min) can be interpreted as a nudging time scale as it dictates how quickly the model approaches the observation.

$W_{xy}$ and $W_z$ define the spatial nudging weight while the temporal weighting function $W_t$ defines the duration and weighting
strength in time. $W_t$ ramps from 0 to 1 and back to 0 (Reen, 2016). The nudging time window and the time between implemented observations was chosen to be 6 hours so that the implemented observations don't overlap each other. This ensures all time steps are nudged while not excessively limiting the model.

Vertical influence was set very small so that observations only affect their own $\eta$ level (Dudhia, 2012). The horizontal weighting factor $W_{xy}$ (see equation: 2 is calculated based on the radius of influence R and the distance between the observation

---

[1]EDDY: HPC cluster at the Carl von Ossietzky Universität Oldenburg, see: https://www.uni-oldenburg.de/fk5/wr/hochleistungsrechnen/hpc-facilities/eddy/

and the grid location $D$. We used the 'Cressman scheme' as the horizontal nudging weighting function with a radius of influence of $R = 180$ km, thereby affecting the whole inner domain.

$$
w_{xy} = \begin{cases} \frac{R^2 - D^2}{R^2 + D^2} & 0 \leq D \leq R \\ 0 & else \end{cases} \tag{2}
$$

## 4 Results

It is important to keep the differences in temporal and spatial resolution between LiDAR measurements and WRF simulation in mind. Furthermore, data availability highly influences the ability to nudge the simulation (see section 2) and compare wind speed statistics.

To quantify the local effect of observation nudging, we investigate the cell closest to the LiDAR measurement location and compare measured and modeled horizontal wind speeds $U$ and direction. Additionally we investigate several sections

at different locations and altitudes within the inner domain to quantify the spatial and temporal impact of single location observation nudging on the entire domain. Vector values of each WRF cell are calculated on the faces of each cell, linearly interpolated to the cell center and rotated from the grid projection to earth coordinate system.

### 4.1 Impact of nudging on wind statistics

Figure 2 shows the scatter plots of measured and simulated horizontal wind speed at various altitudes for times at which

LiDAR data is available. The continuous line represents the linear regression of the data (regression coefficient is displayed in the legend) while the dotted line shows an ideal correlation. The color of the scatter points corresponds to the frequency of occurrence. Multiple wind speed clusters caused by stratification can be identified. While there is a trend towards higher wind speeds with increasing altitude, low wind speeds ($U < 6$ m/s) still occur at high-altitudes. Both simulations overpredict horizontal wind speeds at low-altitudes which is a known problem of WRF and could be attributed to the model not resolving

sub-grid scale roughness elements properly (e.g. modeling strongly simplified parameterization of forests and/or cities) or flaws in the planetary boundary layer model which lead to overly geostrophic winds over land (Mass and Ovens, 2011). Observation nudging improves the overall correlation with measurements at the measurement location as surface influence decays. Both models approach similar values at higher altitudes which could be caused by the lack of observations and therefore observation nudging due to reduced data availability or is indicative of WRF generally being better at modeling more geostrophic winds.

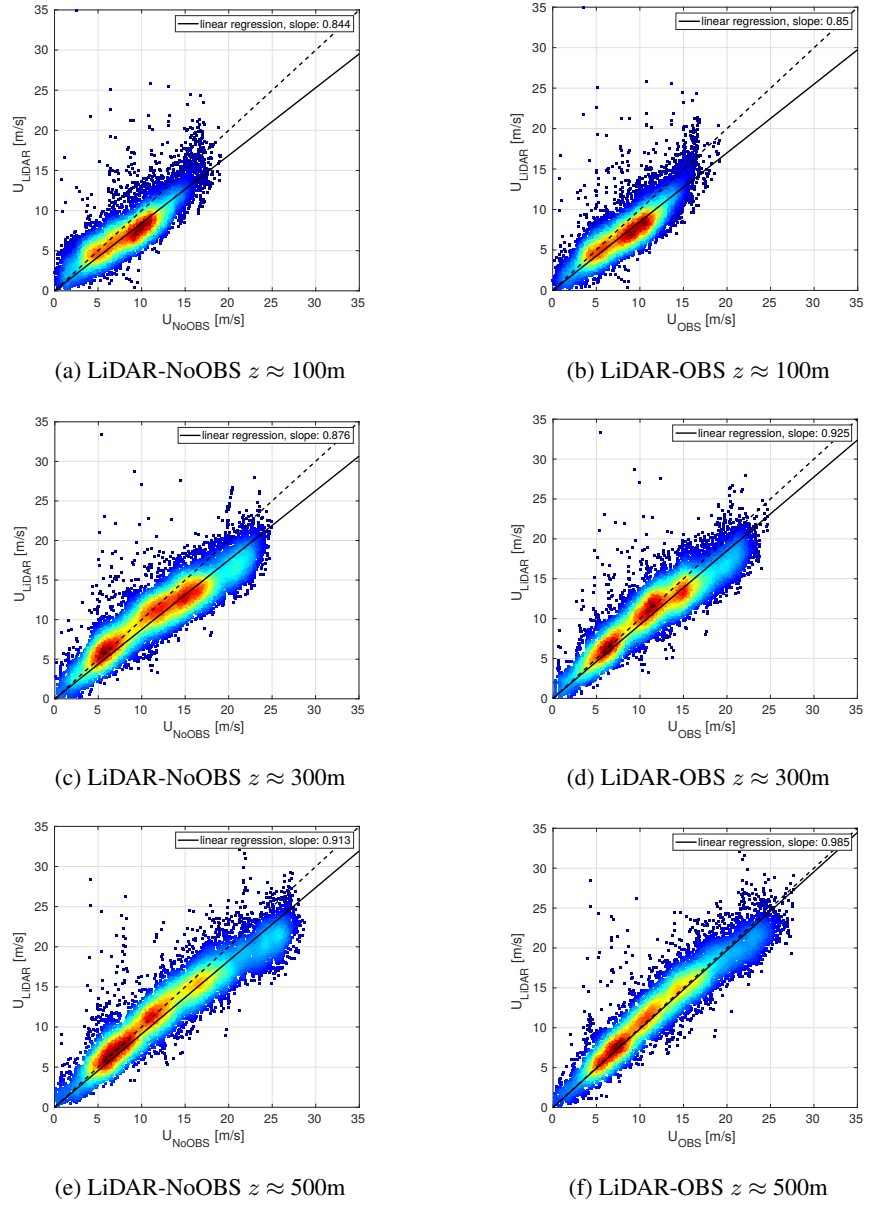

**Figure 2.** Linear Regression of LiDAR-measured wind speeds against NoOBS-modeled (WRF 'baseline run' without observation nudging) wind speeds (left side) and OBS-modeled ('test run' with obsgrid observation nudging) wind speeds (right side), at 100 m (a-b), 300 m (c-d), 500 m (e-f)

The statistical analysis of the absolute difference between the WRF simulated quantities at the measurement location and the LiDAR observations ($\Delta U = U_{WRF} - U_{LiDAR}$; $\Delta\Phi = \Phi_{WRF} - \Phi_{LiDAR}$ wrapped on an interval between $[-\pi, \pi]$) is shown in figure 3 in form of a box plot. The circle corresponds to the median, the colored box indicate the 25th and 75th percentile

and the whiskers to both sides mark $\pm 2.7$ times standard deviation ($\sigma$). Outliers beyond $\pm 2.7\sigma$ are hidden to maintain clarity and readability. The continuous line in the left sub-figure represents the root mean square error (RMSE) between the measured $U_{LiDAR}$ and simulated wind speed $U_{WRF}$.

The simulation with observation nudging (OBS) generally outperforms the unnudged simulation (NoOBS) and is in better agreement with the measurements particularly at altitudes of interest to high-altitude wind energy systems. It furthermore reduces the spread of the bias, illustrated by the smaller whiskers and boxes. The RMSE $\Delta U$ shows similar results for both simulations below 100 m and above 700 m. The largest improvement or smallest error can be found between 300 m and 600 m. This could be explained by a better performance of the mesoscale model at these altitudes due to a reduced impact of the air surface interaction which is strongly parameterized.

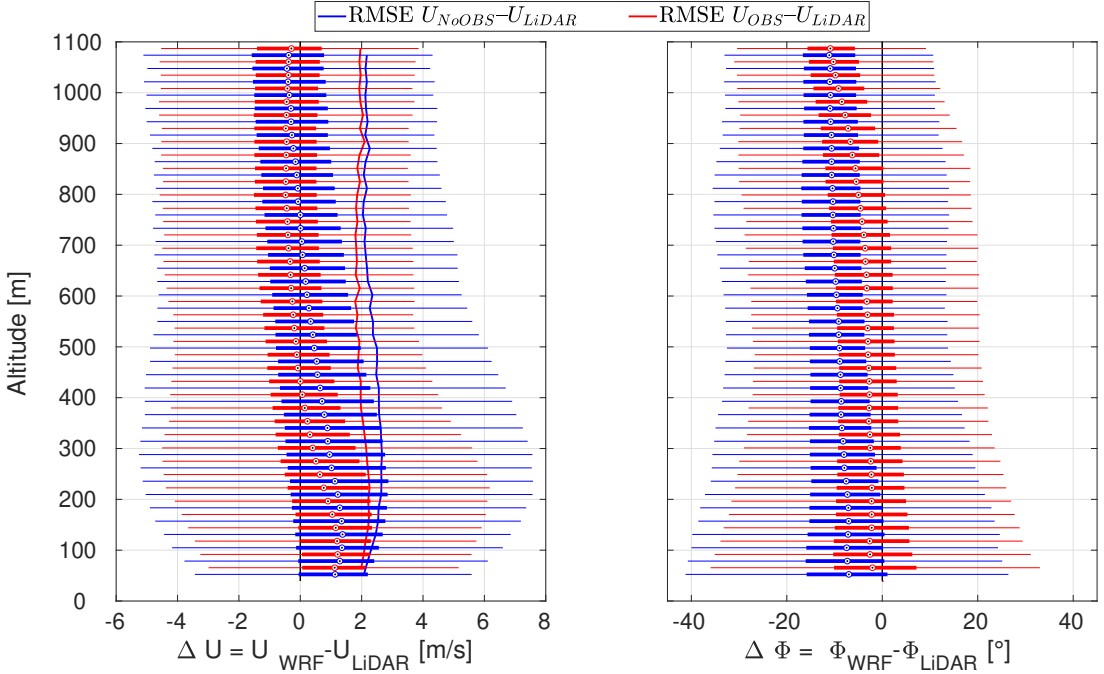

**Figure 3.** Statistical analysis of the bias between simulated and measured wind speed ($\Delta U$) and direction bias ($\Delta \Phi$). The circle corresponds to the median, the colored box indicates the 25th and 75th percentile and the whiskers mark $\pm 2.7\sigma$. The solid lines in the left figure show the RMSE between the modeled and measured wind speed.

The NoOBS shows an almost constant wind direction bias at all altitudes. Observation nudging substantially reduces the directional bias $\Delta \Phi$ up to high-altitudes as can be seen in the right box plot in figure 3. Similar to the wind speed bias, wind direction bias at 1100 m is almost the same for both simulations. The negative wind direction bias represents an anti-clockwise deviation. Other studies (Carvalho et al., 2014; Giannakopoulou and Nhili, 2014) have found similar wind direction biases. A possible reason for this systematic error is that WRF does not adequately resolve surface roughness resulting in lower surface

friction leading to overly geostrophic winds (Mass and Ovens, 2010). The almost constant median wind direction bias indicates that WRF is able to capture the clockwise rotation of the 'Ekman Spiral' in the Northern hemisphere.

## 4.2 Representative nudging results

We compare 10 min mean horizontal wind speed for 24 hours on the 21st of September 2015 in figure 4 to visualize the impact
of observation nudging on the mesoscale model output. White space in the LiDAR measurements (see figure 4a) are data points that have been filtered out due to insufficient data quality. The dashed line is the WRF modeled surface heat flux (SHF) used to estimate atmospheric stability (see sub-section 4.5). The color of the profiles indicate the wind direction and LiDAR measured profiles are shown in grey for comparison. The black dot in each profile marks the altitude of highest wind speed while the black circle indicates the optimal altitude for the operation of an airborne wind energy system based on a simplified power
approximation (see section 4.7). However, the single point representation is only a rough measure of operational altitude since AWES generally sweep a range of altitudes.

Even though observation nudging leads to statistical improvements in wind speed and wind direction prediction over the entire period (compare sub-section 4.1 and 4.4), individual days can still show a decline in model accuracy. The low level jet (LLJ) as well as the high wind speeds at higher altitudes, which the NoOBS model captures fairly well, are significantly weaker
in the OBS model. Implementing additional measurements at a higher frequency might yield results closer to measurements, but adding too many unphysical forcing terms might overly restrict the simulation.

The planetary boundary layer height (PBLH) (black line), which in the MYNN scheme is calculated from the profile of virtual potential temperature and from the profile of the TKE (Brunner et al., 2015; Nakanishi and Niino, 2004), is directly affected by wind speed observation nudging. During the investigated day, observation nudging leads to a lower daytime PBLH.

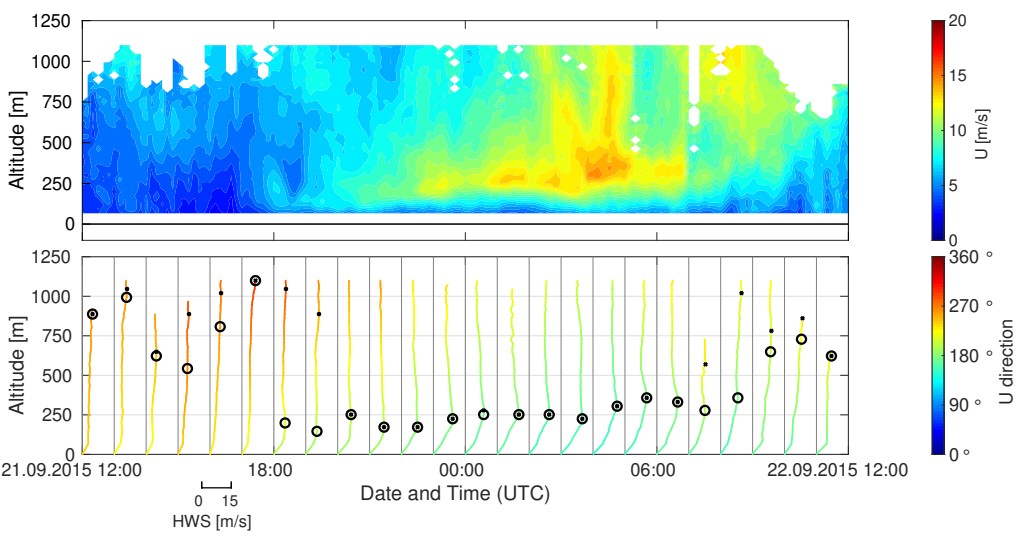

(a) LiDAR

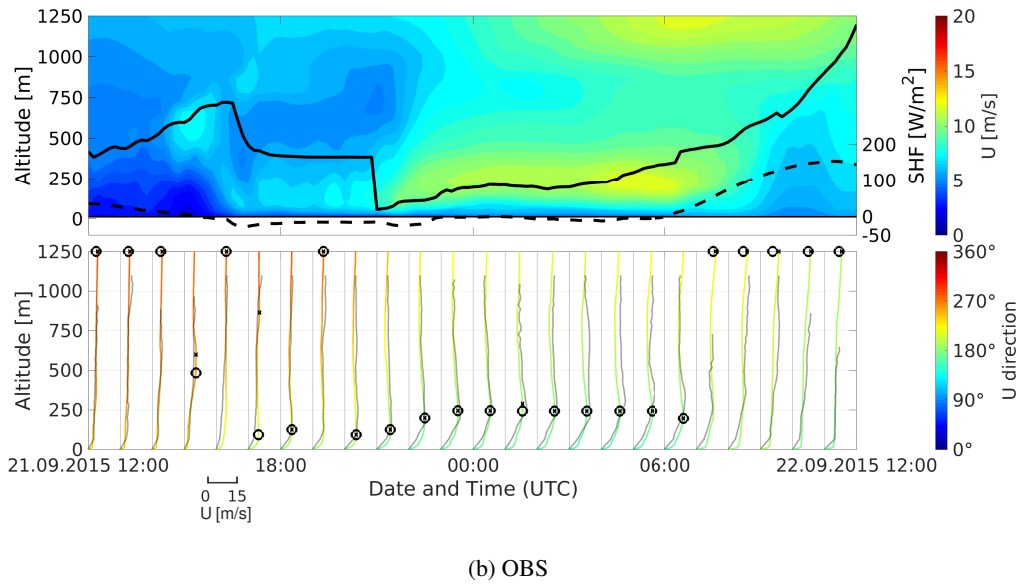

(b) OBS

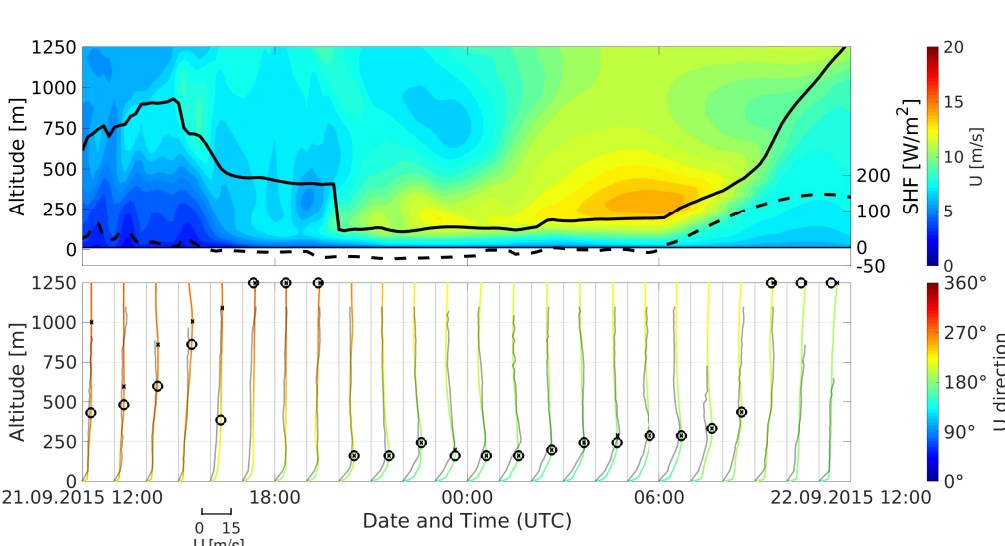

(c) NoOBS

**Figure 4.** Visualization of modeled and measured 10 min mean wind speed, wind direction for 21st September 2015. The respective top figure shows the wind speed and WRF calculated SHF (dashed line). The bottom figure shows each hours 10 min mean wind speed profile colored according to wind direction. X marks the altitude of highest wind speed and ◯ the optimal operating altitude calculated as described in section 4.7

## 4.3 Spatial influence

Single location observation nudging influences the area within the radius of influence ($R_{xy} = 180$ km, see table A1 in the appendix) which here includes the entire inner domain (150 km × 150 km). Figure 5 shows the mean absolute difference of horizontal wind speed ($\Delta U = |U_{OBS}| - |U_{NoOBS}|$) between the OBS and NoOBS model along lines of constant longitude and latitude for the entire simulation period. The grid cell where observations were assimilated is indicated by the vertical line and highlighted by the square marker. The four colors indicate different altitudes. As the outer domains remain unnudged, the boundary conditions of the inner domain remain the same which leads to the rapid decline in absolute difference towards the outside of the domain. The difference in wind speed does not go to exact zero, because results are interpolated to the center of each grid cell. Near surface results close to the measurement location, which is highlighted by the black vertical line, experience the largest change in wind speed (red line, z = 12 m). The asymmetry could be caused by the downstream transportation of nudging effects (dominant wind direction: West).

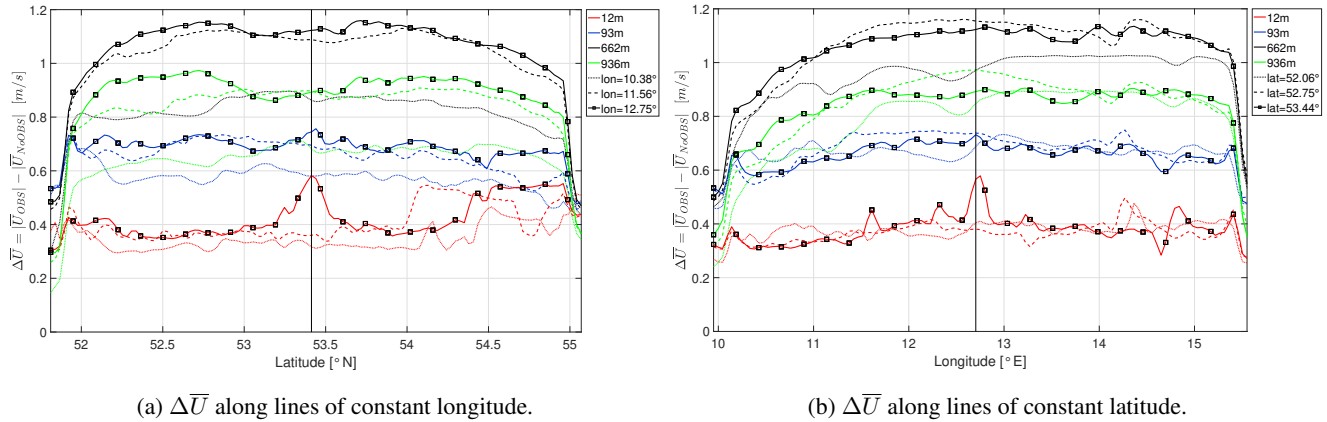

(a) $\Delta\overline{U}$ along lines of constant longitude.

(b) $\Delta\overline{U}$ along lines of constant latitude.

**Figure 5.** Mean absolute wind speed difference $\Delta\overline{U}$ along lines of constant longitude (a) and latitude (b) within the nudged domain. Approximate distance of $d_3 \approx 180$km (dotted lines), $d_2 \approx 75$ km (dashed lines), $d_1 \approx 0$ km (solid line) from the center (Lat: 53° 10' 47.00"N, Lon: 12° 11' 20.98"E) Vertical line highlights the grid cell closest to observation.

## 4.4 Diurnal Variability

Average diurnal variation indicates typical wind speed variations for a given location and period. It further reinforces the benefit of dynamically adapting operating altitudes of AWES. The hourly average LiDAR wind speed depends on data availability described in section 2. LiDAR availability below 100 m on average decreases by about 10 percentage points during the noon hours, while it remains fairly constant at altitudes between 100 m and 300 m. Above this altitude, data availability increases in the afternoon by up to about 15 percentage points (Sommerfeld et al., 2019).

Figure 6a shows the LiDAR measured and mesoscale modeled diurnal wind speed variation at the measurement location filtered by LiDAR availability, i.e. times where no LiDAR data were available were disregarded. A clear diurnal wind speed variation resulting from the cycle of stable and unstable stratification can be identified. On average OBS shows lower hourly

wind speeds than NoOBS and is closer to measurements. The diurnal variation of the 6 months OBS, the 6 months NoOBS the 12 months NoOBS unfiltered data sets (Figure 6b) deviate significantly from the measurements. Observation nudging leads to overall lower wind speeds and wind shear throughout the day in the unfiltered data set. Due to the large difference in average measured and unfiltered modeled diurnal wind speeds, it seems that LiDAR measurements alone can not appropriately represent average wind conditions aloft due to availability bias which also has been observed at other locations (Gryning and Floors, 2019). Therefore, we believe that the nudged data set yields more representative results than the unnudged model or the measurements alone.

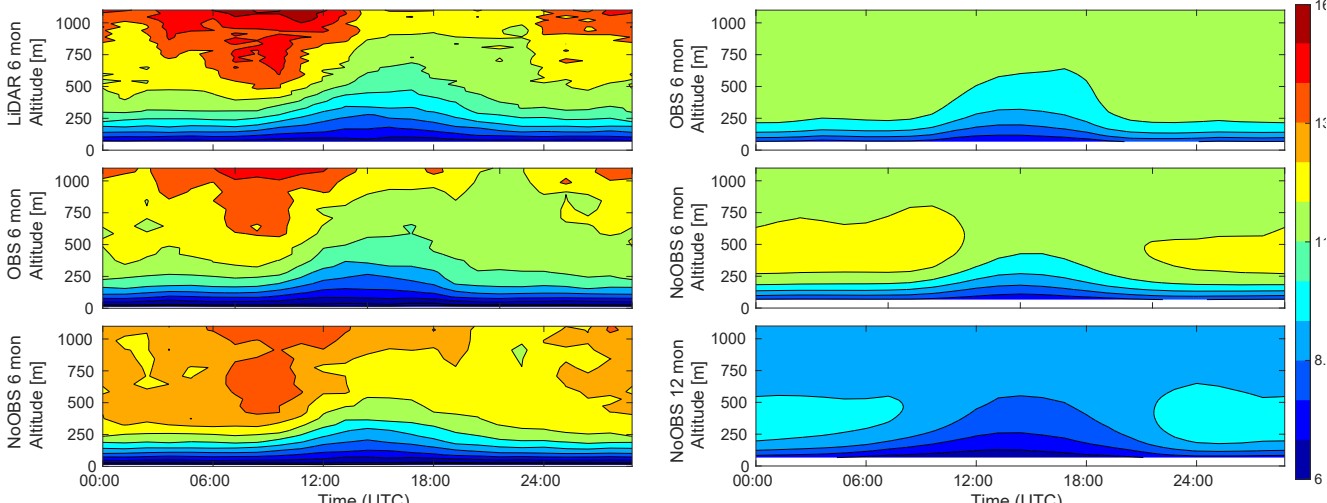

(a) $\overline{U}$ filtered by LiDAR availability: 6 months of LiDAR measurements (top), OBS (center) and NoOBS (bottom) model

(b) $\overline{U}$ unfiltered: 6 months of OBS (top), 6 months of NoOBS (center) and 12 months NoOBS (bottom) model

**Figure 6.** Hourly average diurnal variation of measured and modeled horizontal wind speed $\overline{U}$ filtered by LiDAR availability (a) and unfiltered (b).

## 4.5 Wind speed probability distribution

The common way to approximate the probability distribution of the horizontal wind speed $f(U)$ is the Weibull distribution fit (eq. 3) which describes the statistical distribution as a function of the scale parameter $A$ and the shape parameter $k$ (Troen and Lundtang Petersen, 1989).

$$f_{Weibull}(u) = \frac{k}{A} \left( \frac{u}{A} \right)^{k-1} e^{-\left( \frac{u}{A} \right)^k} \tag{3}$$

Previous investigation of the LiDAR measurements showed a multi-modality in the wind speed frequency of occurrence caused by different atmospheric stability (Sommerfeld et al., 2019). The left column in figure 7 visualizes the entire measured and simulated wind speed frequency distribution. Its corresponding Weibull fit is shown in the center column and the difference

between both can be found on the right hand side. Each row summarizes the various data sets first 6 months LiDAR, then 6 months OBS, 6 months NoOBS followed by 12 months NoOBS.

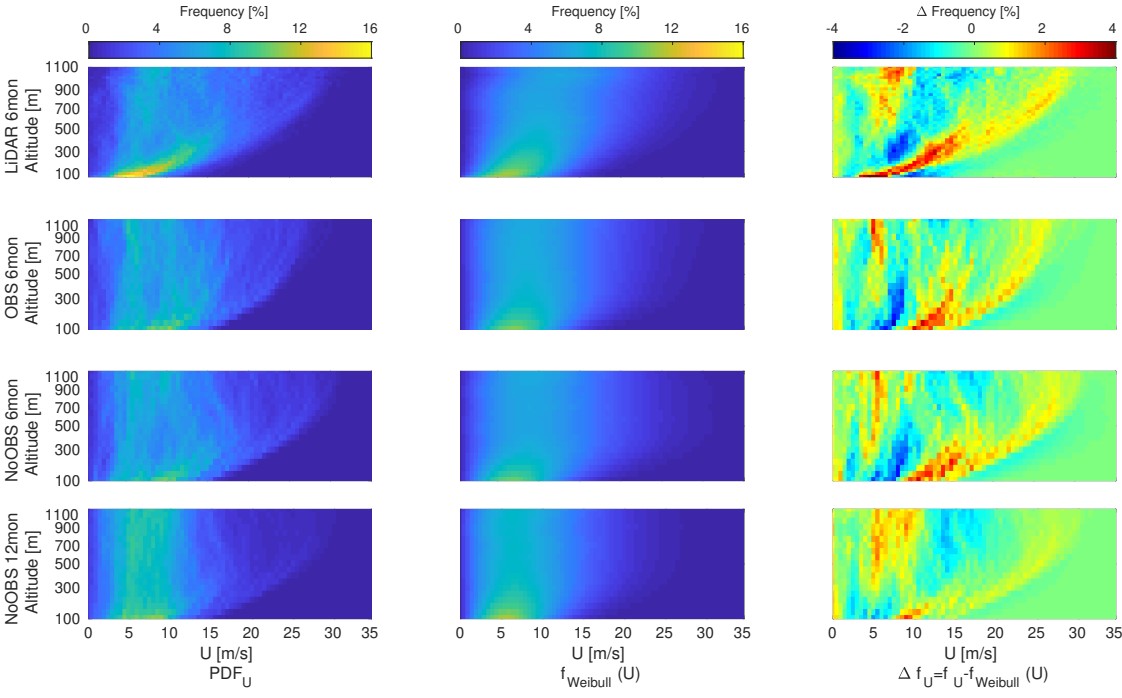

**Figure 7.** Frequency of occurrence (left), Weibull fit (center) and difference between both (right) of 6 months LiDAR measurements (top row), 6 months OBS model (second row), 6 months NoOBS model (third row) and 12 months NoOBS (bottom row). The entire, (not filtered by LiDAR data availability) was used for the WRF data set.

All 6 months data sets show a high occurrence of low and high wind speeds which indicates a multi-modal frequency distribution. This effect is most pronounced in the LiDAR data set. The comparison of wind speed frequency with the Weibull
5   fit (right column) further emphasizes the multi-modality as a simple Weibull fit is not able to capture the higher probability at low and high wind speeds. These distinct flow situations further drift apart with increasing surface-distance. As a result the Weibull distribution overestimates the occurrence of wind speeds in between the two peaks (blue area in right column). Both OBS and NoOBS slightly overestimate low altitude wind speed (see figure 3) compared to LiDAR measurements. Both models and the LiDAR measurements show a broadening of the frequency distribution towards higher altitudes. High wind
10   speeds become more likely while low wind speeds still occur. Therefore, AWES need to be able to operate in a wide range of wind speeds or be controlled in a way that they avoid extreme conditions. The 12 months NoOBS simulation shows lower wind speeds than the 6 months simulations as the included summer months generally have lower wind speeds due to the lower synoptic pressure gradients. The Weibull fit of this simulation tends to overestimate higher wind speeds and underestimate low wind speeds at all altitudes.

Using the sign of the WRF-calculated SHF as a simple proxy to differentiate stable and unstable wind conditions similar to (Sommerfeld et al., 2019). The wind speed distribution follow the expected trends of low wind shear during unstable stratification and higher wind shear and wind speeds during stable stratification (Arya and Holton, 2001). Observation nudging reduces the occurrence of high wind speeds at high-altitudes in comparison to NoOBS and leads to an increase in the probability of wind speeds around 5 m/s during times of positive SHF. The Weibull distribution fit of these sub-states is generally better at representing the modeled wind conditions.

Figure 8 shows the scale parameter $A$, shape parameter $k$ and Hellinger distance $H$ (Upton and Cook, 2008) between the wind speed probability density function (PDF) and the corresponding Weibull distribution fit for LiDAR (1st row), 6 months OBS (2nd row), 6 months NoOBS (3rd row) and 12 months NoOBS (4th row).

The different trends under positive and and negative SHF of both Weibull parameters visualize the existence of entirely different flow regimes. The Hellinger distance between the Weibull fit and PDF (negative SHF: blue and positive SHF: red), the total data and a simple fit (black) as well as between the total data and the weighted sum of both Weibull fits (green) is shown in the right graph. All WRF models show an overall smaller $H$ than a similar analysis of the LiDAR data set (Sommerfeld et al., 2019). The sharp bend in both $A$ and $k$ of the LiDAR data above 750 m is likely caused by insufficient data availability. NoOBS results show a sharp increase of $A$ up to 250 m and a slight reduction above while OBS shows a trend close to the surface, $A$ values remain almost constant above 500 m. No data set shows a convergence of $A$ at higher altitudes indicating that these wind conditions are driven by different conditions in the free atmosphere. 12 months NoOBS simulations show lower scale parameter values as they include generally slower winds during summer. While $A$ trends are quite different for LiDAR and WRF, $k$ trends are more similar. They peak between 150 and 250 m and are especially high during stable stratification (Monahan et al., 2011). OBS trends of $k$ are generally closer to measurement results than NoOBS.

Even though the Hellinger distance of individual Weibull fits for times of positive or negative SHF is generally higher than the Weibull fit of the entire data set, the weighted sum of both individual fits yields the best result at all altitudes. The 12 months Weibull fit using the entire data set performs comparable to weighted sum up to an altitude of about 250 m.

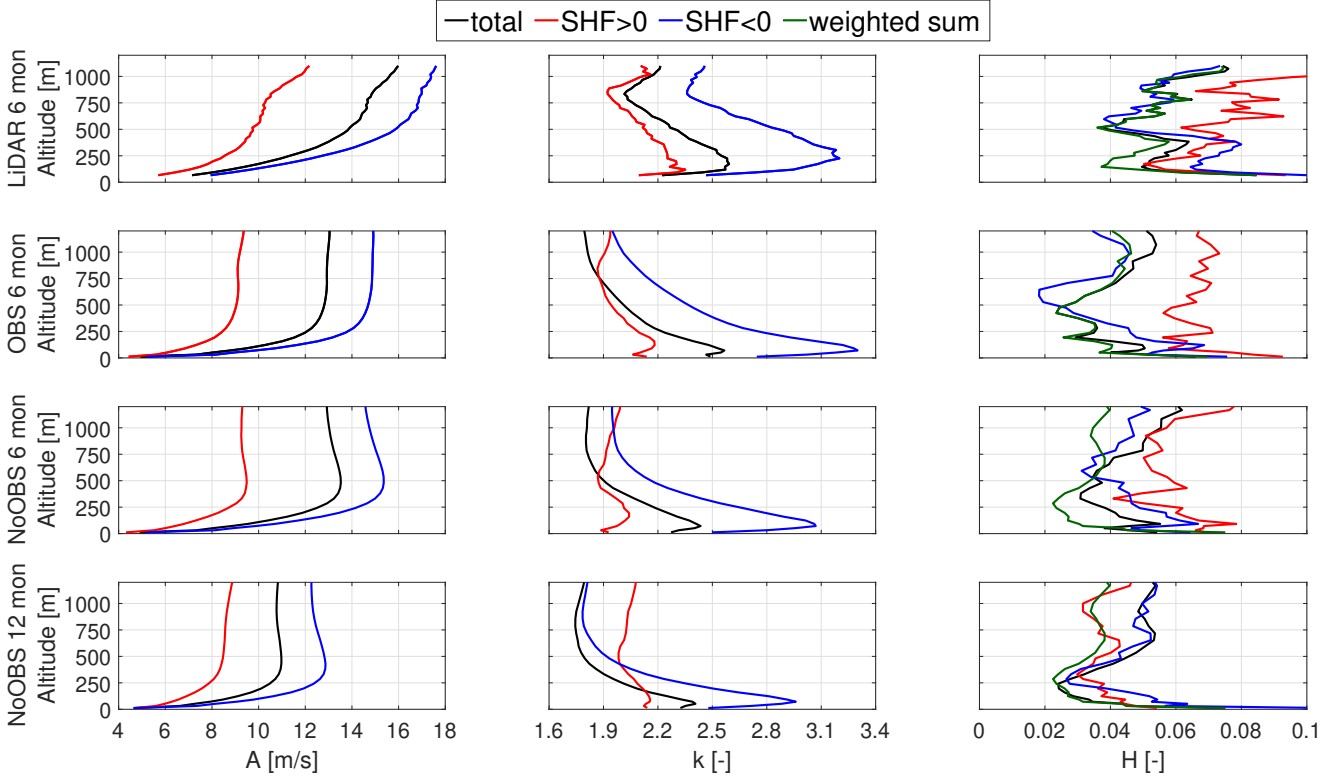

**Figure 8.** Weibull parameter trends over altitude and goodness of fit quantified by the Hellinger distance (right) over altitude for 6 months of LiDAR measurements (1st row), the 6 months OBS model (2nd row), 6 months NoOBS model (3rd row) and the 12 months NoOBS model (4th row)

## 4.6 Effect of stability on average wind shear

Atmospheric stability highly influences the shape of wind speed profiles which is important for determining optimal operating conditions for AWES (see section 4.7). Obukhov length $L$ (Obukhov, 1971; Sempreviva and Gryning, 1996) is commonly used to categorize the stability of the boundary layer. Here the application is extended to mid-altitudes. $L$ is defined by the simulated friction velocity $u_*$, virtual potential temperature $\theta_v$, potential temperature $\theta$, kinematic virtual sensible surface heat flux $Q_S$, kinematic virtual latent heat flux $Q_L$, the von Kármán constant $k$ and gravitational acceleration $g$. Table 1 summarizes the frequency of occurrence of each stability class.

$$L = \left( \frac{-u_*^3 \theta_v}{kg} \right) \left( \frac{1}{Q_S} + \frac{0.61}{Q_L \theta} \right) \tag{4}$$

**Table 1.** Stability classes according to Obukhov length calculated based on WRF results (Floors et al., 2011)

| Stability classes | L [m] | OBS 6 mon | NoOBS 6 mon | NoOBS 12 mon |
|---|---|---|---|---|
| Unstable (u) | $-200 \leq L \leq -100$ | 5.69 % | 3.93 % | 7.27% |
| Near unstable (nu) | $-500 \leq L \leq -200$ | 8.21 % | 6.35 % | 7.09 % |
| Neutral (n) | $\|L\| \geq 500$ | 28.71 % | 29.76 % | 20.71 % |
| Near stable (ns) | $200 \leq L \leq 500$ | 18.26 % | 19.30 % | 12.56 % |
| Stable (s) | $50 \leq L \leq 200$ | 18.63 % | 18.6 % | 17.24 % |
| Very stable (vs) | $10 \leq L \leq 50$ | 6.15 % | 6.75 % | 10.04% |
| Other | $-100 \leq L \leq 10$ | 14.76 % | 15.31 % | 25.09 % |

In comparison with the unnudged simulation, OBS shows an increase in unstable and near unstable situations. Stable and near stable stratification seems almost unaffected by OBS nudging, while neutral and very stable stratification occur slightly less often. This might improve the overall predicting capabilities of WRF as the MYNN 2.5 boundary layer scheme overestimates the frequency of very stable conditions with an error of up to 9 % (Krogsæter and Reuder, 2015). Neutral conditions, still commonly used in many wind energy siting applications, only occur about 30 % of the time during the measurement period and only about 20 % of the time during the one year reference NoOBS simulation.

Figures 9 shows the frequency distribution of the different stability categories for each with the mean highlighted by white squares. All categorize show distinct trends and distributions that are consistent between data sets, which contribute to the multi-modality of the overall wind speed frequency distribution. The difference in high-altitude wind speeds between stratifications indicate the influence of different geostrophic wind conditions. The categorization by $L$ is based on surface data and seems to be valid within the lower part of the atmosphere where the spread of the corresponding frequency distribution is relatively small in comparison to high altitudes. This is particularly true for stable and neutral stratification where wind speeds above approximately 200 m spread widely. Unstable conditions are probably more consistent because of increased mixing from the surface up to high altitudes. The divergence of wind speeds towards higher altitudes indicate inhomogeneous atmospheric stability and suggests that surface-based stability categorization is insufficient for higher altitudes. Wind speed extrapolation based on low altitude measurements can lead to a misestimation of mid-altitude wind conditions, especially during neutral and stable conditions close to surface. (Konow, 2015)

Altitudes below 200 m are least affected by observation nudging as OBS remains almost unchanged from NoOBS (see section 4.1). Stable profiles show a peak at around 300 m which is indicative of a characteristic low level jet. Comparing OBS and NoOBS 6 months, observation nudging seems to reduce the spread at higher altitudes within each category except very

stable. The impact of observation nudging on wind profiles during unstable stratification is relatively low while wind speed profiles under neutral and stable stratification are more affected.

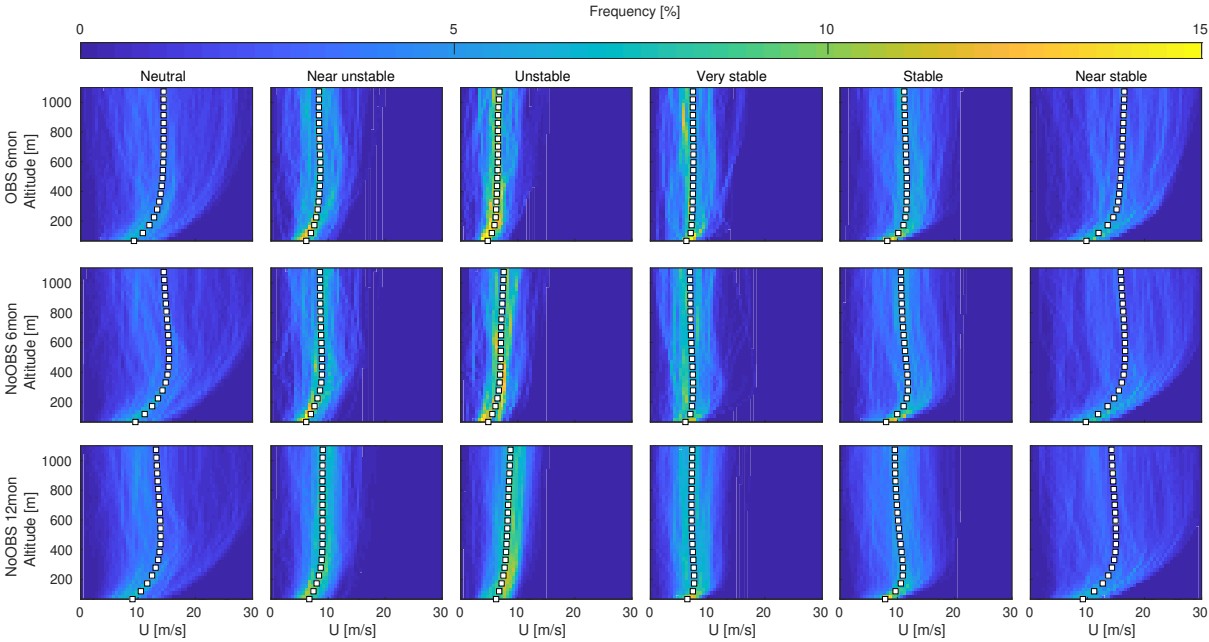

**Figure 9.** Wind speed $U$ frequency of occurrence and mean (white square) categorized by atmospheric stability according to Obukhov length $L$ (see Table: 1) for 6 months OBS (top), 6 months NoOBS (center) and 12 months NoOBS (bottom).

### 4.7 Optimal operating altitude and power production

We estimate optimal operating altitude and traction power of a ground-generator AWES using a simple ground-generator
5 (pumping-mode) AWES point-mass model adapted from (Schmehl et al., 2013). We focus on 6 months OBS as we previously proved increased accuracy and use 12 months NoOBS to estimate annual values. The estimated optimal power per unit lifting area of the wing $p_{opt}$ is described by:

$$p_{opt} = \frac{\rho_{air}}{2} U^3 \sqrt{c_L^2 + c_D^2} \left[1 + \left(\frac{c_L}{c_D}\right)^2\right] f_{opt} \left(\cos\varepsilon \cos\phi - f_{opt}\right)^2 = \frac{2}{27}\rho_{air} U^3 \sqrt{c_L^2 + c_D^2} \left[1 + \left(\frac{c_L}{c_D}\right)^2\right] \cos\varepsilon^3 \quad (5)$$

Air density $\rho_{air}$ is calculated by a linear approximation of the standard atmosphere (ISO 2533:1975) ($\rho_{air}(z) = 1.225 -$
10 $0.00011z$ [kgm$^{-3}$]. Losses associated with mispositioning of the aircraft relative to the wind direction, expressed by azimuth angle $\phi$ and elevation angle $\varepsilon$ relative to the ground station, are included in the model. Additional losses caused by gravity, tether sagging and tether drag are neglected. As a result, lift $F_L$ and drag $F_D$ force and therefore lift ($c_L =$1.7) and drag coefficient ($c_D =$0.06), which are assumed to be constant, are geometrically related to the apparent wind velocity. Assuming an optimal tether speed and a quasi-steady state with the wing moving directly cross-wind with a zero azimuth angle ($\phi = 0$)

relative to the wind direction we can estimate the optimal traction power. Optimal elevation angle ($\varepsilon_{opt}$) and operating altitude ($z_{opt}$) are geometrically related to the assumed to be constant tether length ($l_{tether}$) ($\sin \varepsilon_{opt} = \frac{z_{opt}}{l_{tether}}$).

Figure 10 summarizes the frequency of optimal operating altitude and optimal power assuming a constant tether length of 1500 m. The white solid line shows the cumulative frequency of optimal operating altitude. Both simulations for this particular
5   location and time period show similar trends with the most probable optimal altitude between approximately 200 and 400 m. Times of very high traction power are fairly rare and likely associated with low level jets. Lower power at higher altitudes is caused by the misalignment losses.

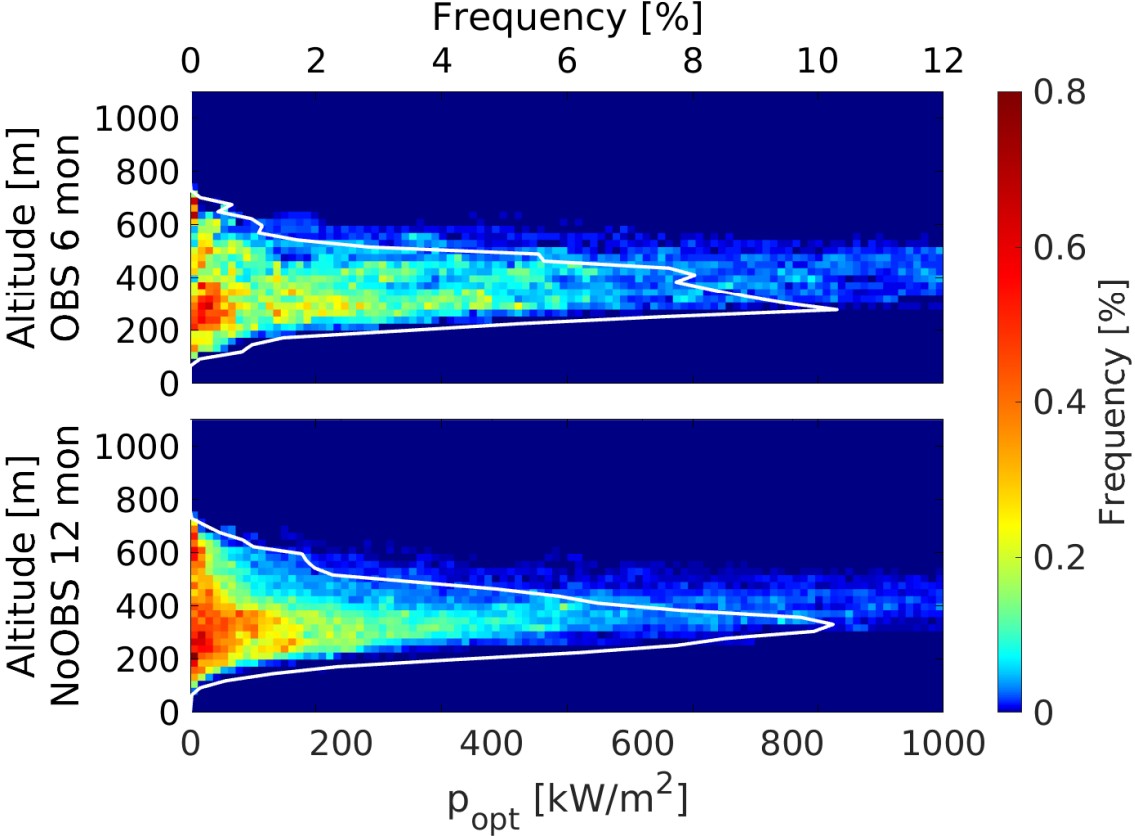

**Figure 10.** Frequency of optimal traction power over optimal operating altitude based on 6 months OBS (top) and 12 months NoOBS (bottom) assuming a constant tether length of 1500 m. The continuous white line shows the frequency of optimal operating altitude for the whole power range (top abscissa axis).

Figure 11 estimates the optimal traction power and operating altitude as a function of tether length based on the mean wind speed profile of atmospheric stability condition (figure: 9). The tether length of each estimation is assumed to be constant and
10   used to calculate the optimal elevation angle. The axis limits of different atmospheric conditions had to be adjusted as the calculated power varied in order of magnitudes. All estimates show diminishing benefits of a longer tether. These incremental

gains would probably be negated by additional drag and weight associated losses. Winds during times of very stable and unstable stratification lead to a clear optimal altitude independent of tether length between 200 and 400 m while weakly stable and shear-driven wind speed profiles lead to higher optimal operating altitudes and a broader range of optimal altitudes as a function of tether length.

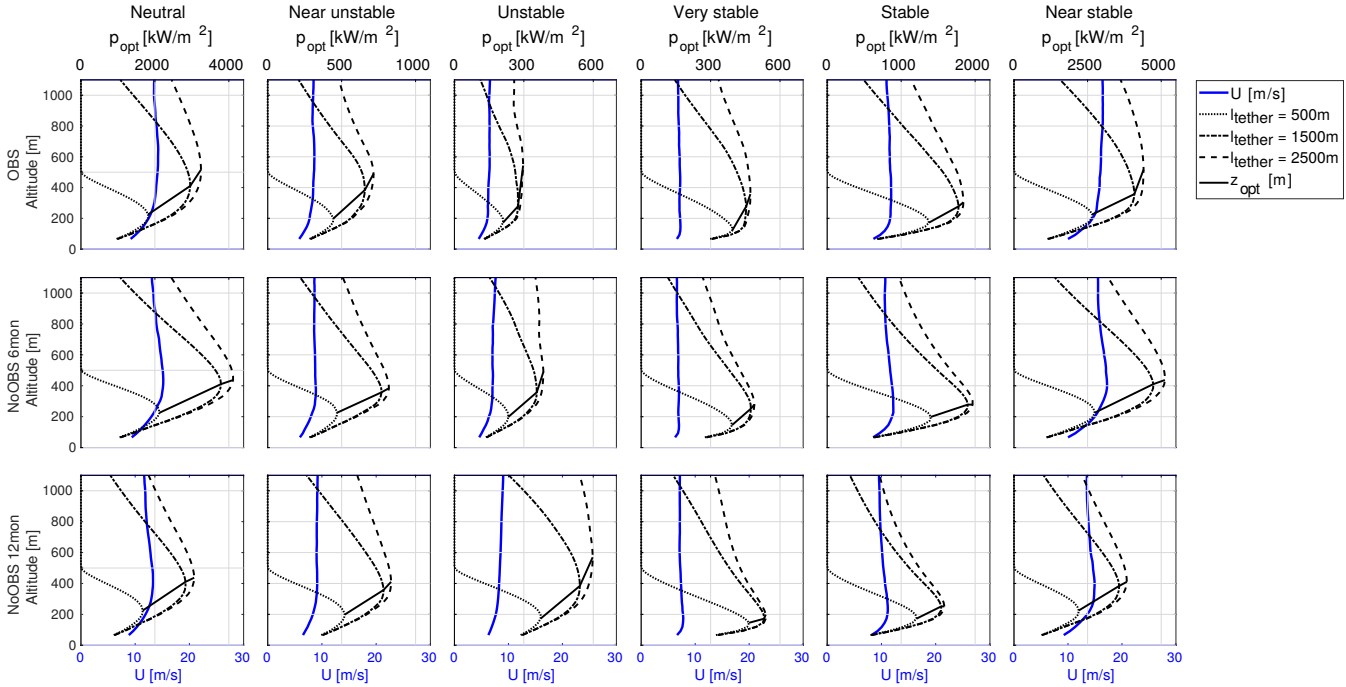

**Figure 11.** Optimal traction power per wing area $p_{opt}$ (dashed lines) and optimal operational altitude (solid line) estimated based on mean wind speed profiles categorized by Obukhov length ($L$) for 6 months OBS, 6 months NoOBS and 12 months NoOBS simulation with varying tether length ($l_{tether}$ = 500 - 2500m).

## 5    Conclusion

Six months of LiDAR measurements up to 1100 m were assimilated in to a mesoscale model using observation nudging. An unnudged reference model (NoOBS), the nudged model (OBS) outputs and LiDAR measurements were compared in terms of wind speed and direction statistics, wind profile shape at the measurement site as well as spatial differences were quantified. Observation nudging only has marginal impact on simulated surface layer wind speeds as ground effects dominate the WRF model. Wind speeds between 300 and 500 m were most affected by observation nudging. Modeled wind speeds at these altitudes are statistically closest to measurements, making this an adequate approach for resource assessment of mid-altitudes as measurement availability decreases. The impact of nudging weakens above these altitudes. Whether this is caused by lower measurement data availability or a generally better performance of the mesoscale model above the surface layer could not be determined. Observation nudging reduced the seemingly systematic wind direction bias between simulation and measurements

at all altitudes. Due to the lack of high-resolution measurements at high-altitudes, unnudged mesoscale model data present the best we have got in terms of preliminary resource assessment.

Filtering the mesoscale model data according to LiDAR data availability yields similar diurnal variation with OBS being closer to measurements. Comparing the diurnal variation of the unfiltered model wind speeds to measurements shows a significant deviation which is likely caused by insufficient LiDAR data availability at higher altitudes. The bias between real and LiDAR measured wind speed, which depends on the applied CNR threshold and data availability, can result in a misrepresentation of the actual wind conditions especially at higher altitudes. Mesoscale models, particularly with observation nudging, can be used to account for this error. LiDAR measurements seem to be biased towards high wind speeds as measured winds are generally higher than the unfiltered mesoscale model data. The impact of observation nudging on the wind profiles in case of an unstably stratified boundary layer is relatively low while wind speed profiles under stable stratification are significantly affected. At the measurement location OBS is overall closer to measurements especially between 200 and 600 m. Variations of stratification, primarily those associated with the diurnal cycle, lead to a multi-modal wind speed frequency distribution which is better represented by the weighted sum of two Weibull fits than by a single Weibull fit. Obukhov length categorized wind speed profiles, especially during neutral and stable conditions close to surface, show a divergence with height. This indicates inhomogeneous atmospheric stability and suggests that surface-based stability categorization is insufficient for higher altitudes.

Optimal AWES operating altitudes and power output per wing area were estimated based on a simplified model for six months of OBS and twelve months of NoOBS. The model neglects kite and tether weight as well as tether drag. Accounting for these losses, which are proportional to tether length, will reduce the performance of the AWES. Results for both wind speed data sets show the highest potential at an altitude between 200 and 600 m above which the losses associated with the elevation angle are too high. A comparison of different tether lengths under average wind speeds associated with different atmospheric stability conditions show diminishing returns in terms of power output for tether lengths longer than 1500 m. While higher altitudes can be potentially be reached, optimal operating altitude remains almost unchanged. The highest energy potential and operating altitude is associated neutral and stable stratification. Unstable conditions result in significantly lower energy potential due to lower, almost altitude independent average wind speeds.

Future studies include using the enhanced mesoscale model output to drive large-eddy simulations, to provide a better insight into mid-altitude turbulence. The resulting data set will lead to the development of a mid-altitude engineering wind model which can be used for design, load estimation, control and optimization of Airborne Wind Energy Systems. Mesoscale model data will be implemented into an AWES optimization framework to quantify the impact of various wind speed profiles on power production, optimal trajectory and system size. Furthermore, the possibility of merging the mesoscale output with LiDAR measurements to fill gaps in the measurement data set to reduce the wind speed bias introduced by LiDAR availability is being investigated.

# 6 Appendix

**Table A1.** Namelist parameters for WRF 3.6.1 observation nudging

| WRF input parameter | value | WRF input parameter | value |
|---|---|---|---|
| grid_fdda | 1,1,1, | Cressman Scheme | 1 |
| gfdda_inname | "wrffdda_d<domain>", | time_step | 60 |
| gfdda_end_h | 99999, 99999, 99999, | obs_rinxy | 240,240,180 |
| gfdda_interval_m | 360, 360, 360, | obs_rinsig | 0.1 |
| fgdt | 0, 0, 0, | obs_twindo | 3, 3,3 |
| if_no_pbl_nudging_uv | 0, 0, 1, | auxinput11_interval_s | 360, 360, 360 |
| if_no_pbl_nudging_t | 0, 0, 1, | obs_dtramp | 40 |
| if_no_pbl_nudging_q | 0, 0, 1, | obs_nudge_wind | 1,1,1 |
| if_zfac_uv | 0, 0, 0, | obs_coef_wind | 6.E-4,6.E-4,6.E-4 |
| k_zfac_uv | 0, 0, 30, | iobs_onf | 2,2,2 |
| if_zfac_t | 0, 0, 0, | auxinput11_interval_s | 360, 360, 360 |
| k_zfac_t | 0, 0, 30, | auxinput11_end_h | 6, 6, 6 |
| if_zfac_q | 0, 0, 0, | if_no_pbl_nudging_uv | 0, 0, 1 |
| k_zfac_q | 0, 0, 30, | if_zfac_uv (max_dom) | 0,0,30 |
| guv | 0.0003, 0.0003, 0.0003, | sf_sfclay_physics | 5, 5, 5 |
| gt | 0.0003, 0.0003, 0.0003, | sf_surface_physics | 4, 4, 4 |
| gq | 0.0003, 0.0003, 0.0003, | bl_pbl_physics (max_dom) | 5, 5, 5 |
| if_ramping | 1, | bl_mynn_tkeadvect | .true.,.true.,.true. |
| dtramp_min | 60.0, | ra_lw_physics | 1 ,1,1 |
| io_form_gfdda | 2, | ra_sw_physics | 1 ,1,1 |
| obs_nudge_opt | 0,0,1 | mp_physics | 5, 5, 5 |

*Competing interests.* The corresponding author (Markus Sommerfeld) confirms on behalf of all authors that there have been no involvements that might raise the question of bias in the work reported or in the conclusions, implications, or opinions stated.

*Acknowledgements.* The authors thank the federal ministry for economic affairs and energy for funding of the "OnKites I" and "OnKites II" project [grant number 0325394A] on the basis of a decision by the German Bundestag and project management Projektträger Jülich. The simulations were performed at the HPC Cluster EDDY, located at the University of Oldenburg (Germany) and funded by the federal ministry for economic affairs and energy under grant number 0324005. We thank the PICS and the DAAD for their funding. We further thank all the technicians and staff at IWES for carrying out the measurement campaign at Pritzwalk and their support in evaluating the data.

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
