# Peer review of "Improving mesoscale wind speed forecasts using LiDAR-based observation nudging for airborne wind energy systems"

_Wind Energy Science, 2019_

## Short Comment (SC1) · 10 Apr 2019

We have observed a large number of scientific and patent publications focused on high-altitude wind exploitation, reflecting a truly exponential trend. KiteGen, as the first global entity to produce energy using this revolutionary method, finds itself in a difficult position following the massive amount of material produced by third parties and the consequential technical inaccuracies desperately needing rectification. The latter has a detrimental effect on the potential acceptance of the concept and occasionally leads to technological nonsense, weakening the potential for widespread

common awareness of this powerful technology that has the potential to enable global transition from fossil fuel energy sources. We have observed that, due to the absolute originality and novelty of this concept, there is a lack of qualified peer review, and blatant errors have been propagated and transferred, undisturbed, from one poorly informed publication to another, with no-one critically re-analyzing their stratified assumptions. We have also observed that these same errors have confused the informal competition that has grown over time around our project, among what seems a hundred actors, leading to the copious physical development of low TPL and/or unfeasible or extremely deficient alternative architectures. KiteGen has long refrained from scientific communication due to the absolute certainty of our original and long-established architectural and scientific consistency, but having the devil hiding in the details of the technological issues, this certainty has correctly governed and become involved daily in our developmental activities. The subject paper, despite the voluminous data and formal processes involved, is an example of a misguided effort that fails to produce significant forward progress in this scientific and technological domain and risks becoming completely out of sync and out of the dynamic range of most of the architectures and technology cited in block by the article. We hope that our position will be widely accepted through reading and understanding the comments we make available in the attached paper, accompanied by the appreciation of the articulation of this logical, albeit rare, thinking in professional and strategic energy planning.

Please also note the supplement to this comment:
https://www.wind-energ-sci-discuss.net/wes-2019-7/wes-2019-7-SC1-supplement.pdf
* * *
**Fig. 1.**

**Supplement:**

[Figure]

Interactive Commentary on "*Improving mid-altitude mesoscale wind speed forecasts using LiDAR-based observation nudging for Airborne Wind Energy Systems*"

Markus Sommerfeld1, Curran Crawford1, Gerald Steinfeld2, and Martin Dörenkämper

(2019)  DOI: 10.5194/wes-2019-7

https://www.wind-energ-sci-discuss.net/wes-2019-7/wes-2019-7.pdf

Author: Massimo Ippolito

Institution: KiteGen Research

Author's Note

Former Director of Respira Interdepartmental Laboratory of Polytechnic of Torino

Founder and CEO of KiteGen Research srl

Correspondence to: m.ippolito@kitegen.com

**Table of content**

**Abstract**

We have observed a large number of scientific and patent publications focused on high-altitude wind exploitation, reflecting a truly exponential trend. KiteGen, as the first global entity to produce energy using this revolutionary method, finds itself in a difficult position following the massive amount of material produced by third parties and the consequential technical inaccuracies desperately needing rectification. The latter has a detrimental effect on the potential acceptance of the concept and occasionally leads to technological nonsense, weakening the potential for widespread common awareness of this powerful technology that has the potential to enable global transition from fossil fuel energy sources. We have observed that, due to the absolute originality and novelty of this concept, there is a lack of qualified peer review, and blatant errors have been propagated and transferred, undisturbed, from one poorly informed publication to another, with no-one critically re-analyzing their stratified assumptions. We have also observed that these same errors have confused the informal competition that has grown over time around our project, among what seems a hundred actors, leading to the copious physical development of low TPL[1] and/or unfeasible or extremely deficient alternative architectures.

KiteGen has long refrained from scientific communication due to the  absolute certainty of our original and long-established architectural and scientific consistency, but having the devil hiding in the details of the technological issues, this certainty has correctly governed and become involved daily in our developmental activities.

If it could be agreed as true that KiteGen is now the premier exercise in applied technology and good engineering practices, it would ensure the large-scale production of generating equipment and the resulting sustainable energy therefrom. Obviously, we do not claim that everything has been perfected; further improvements are probable and desirable, *but* in classic Pareto progression. The subject paper, despite the voluminous data and formal processes involved, is an example of a misguided effort that fails to produce significant forward progress in this scientific and technological domain and risks becoming completely out of sync and out of the dynamic range of most of the architectures and technology cited in block by the article.  We hope that our position will be widely accepted through reading and understanding the comments we make available in this paper, accompanied by the appreciation of the articulation of this logical, albeit rare, thinking in professional and strategic energy planning.

*Keywords:* Lidar, Sodar, troposphere, altitude, wind, KiteGen, electrical, energy, baseload, storage, HAWES, Capacity Factor.
* * *
[1] Technology Performance Level, the most useful parameter assessing and comparing new energy concepts.

Interactive Commentary on "Improving Mid-altitude Mesoscale Wind Speed Forecasts Using

LiDAR-based observation nudging for Airborne Wind Energy Systems"

**Method**

The article is certainly of good quality, providing precious original data and measurements, that can be profitably added to the collection of previous work in general to state definitively and positively that high-altitude wind must be considered a new and inexhaustible source of energy that also totally solves the energy storage issue. However, the wind turbine-minded mindset of the authors should be noted, which emphasizes wind stability classes rather than the opportunity to increase the capacity factor by exploiting the freedom choose the various operational heights of tropospheric wind. While the site's location regarding wind stability is important for wind turbines, not only for productivity but also for structural resistance evaluation, it is completely negligible for effective High Altitude Wind Energy System (HAWES) architectures where dynamic lightness and swiftly responsive numerical control allow for immediate feedback and adaptive strategies. Furthermore, the authors' over-simplified model of HAWES leads to inappropriate recommendations and the invalid/improper use of this valuable data-set, resulting in several orders of magnitude out of the dynamic of the devices[4], resulting in unfortunate and useless, though sophisticated, elaboration of their findings. The difference of the article's evaluations from reality is so great that it prevents us from going further in any analysis; in the meantime recommending giving up on data massaging or model nudging, at least in the HAWES domain.

As a suggestion for further investigation that your skilled team may conduct, we believe that your work would be very useful in validating the assumptions made by Archer C., Caldeira K. (2009) [3], at mid-altitude mesoscale, regarding competitive or collaborative comparisons between the array of traditional systems, and the opportunities to exploit the naturally-stored energy in geostrophic wind by means of High Altitude Wind Energy Systems. LiDAR-based observations from two or more geographically-distant locations would assess the capability of the winds in the troposphere to provide at least minimally-required power 99.9% of the time when two appropriately-distanced power harvesting devices are interconnected and simultaneously utilized as one.

**High-Altitude Wind Equipment, Design And Architectures.**

Text quoted from page 2 paragraph 5:

*"Unlike conventional wind energy which has converged to a single concept with three blades and conical tower, several different AWES designs are under investigation by numerous companies and research institutes worldwide (Cherubini et al., 2015). Various concepts from*

*ring shaped aerostats, to rigid wings and soft kites with different sizes, rated power and altitude ranges compete for entry into the marketplace. Since this technology is still in an early stage, none are commercially available.”*

Kitegen financed this (Cherubini et al. 2015) [2] study, as reported in the acknowledgments of the same, without exerting any pressure on the authors regarding the content, in the conviction it would enable the reader to make an independent comparison among the different technology proposals. To any skilled technician in the art capable to evaluate, the differences and various advantages of the architecture proposed by KiteGen are blatantly clear. We have already experienced several learned opinions that confirm KiteGen has not only the best architecture, but also the only one feasible on an industrial scale while simultaneously fulfilling grid requirements. Other available meta-indicators show an overwhelming convergence toward the concepts patented by KiteGen, such as the number of new patent applications replicating our approach, and the numerous independent scientific publications addressing our architecture. Surprisingly, the requisite technological skills needed to guide and discern among alternatives don't seem widely evident among academia and researchers, even in works/papers like this one where our interactive commentary is addressed despite revealing volumes of good data and astute formal processes.

KiteGen, as a pioneer, first in producing energy with this new concept, fully patented, and first completing the research, including a 10-year continuous assessment of the architectures, cannot anymore accept, without some reaction, the informal competition around our proprietary technology and the misguided "information" delaying its understanding and acceptance .

From (Cherubini et al. 2005), the classification of flygen and groundgen is clear, as the different wings or wind-harnessing devices, propeller adopted or different rotokite concepts.

The inadequate buoyancy of helium-filled blimps, a few N per cubic meter unit, cannot withstand the very high horizontal force of the wind, correctly reported and highlighted in the article, which implies the altitude cannot be arbitrarily chosen by the control of such devices. Other dysfunctions will occur for rigid flat wings that cannot structurally withstand the forces without unacceptable weight (longeron beam) or flexible fabric wings that cannot maintain sufficient aerodynamic efficiency or survive the forces and aerodynamic stress required for energy production. In particular, there is the recent demonstration corresponding to the conclusion of the applied research conducted by KiteGen, which delivers the complete and validated projection of generators on an industrial scale, which will put an end to architectural speculation, as this development demonstrates adherence to the best possible specifications in terms of LCA and energy quality, outperforming the matured wind turbines both in onshore and offshore applications by a thousand times.

It seems impossible to avoid the (*false, but overwhelming*) politically-correct urge to set aside/ignore the comparisons or architecture adoption to the focus on the most promising concept. There remains the need to stress, in any case, the  introduction of robust scientific

criteria, stated with certainty and generally requiring that such an analysis must be specifically tailored to each of the architectures and harnessing strategies depicted, instead of generically considering the domain as a whole. Each architecture has different requirements and behavior within wind forces and speeds and different capacity factors. For example, the paper's authors deduced and recommended an ideal and quite precise operating altitude of 200-400 m with a computed wing of 28 of gliding factor. Such a wing, in a pumping kite architecture, will fly in crosswinds at speeds of 100m/s and over. The suggested altitude provides a very tight cone of operation, which would require the wing to maintain an excessively tight direction-changing path in airspace and an impractical few seconds to complete a stroke inside the lemniscate (the characteristic "eight"-shaped path).

**Accurate Data of High Altitude Wind; Is It required?**

Text quoted from page 2 paragraph 10:

*"Developers and operators of large conventional wind turbines, AWES and drones require accurate wind data to estimate power and mechanical loads."*

This statement is absolutely wrong, and risks concealing one of the primary architectural advantages of KiteGen. The KiteGen design concept is totally different because it does not have fixed structures that have to withstand the worst weather conditions that occur once in decades. It starts from an arbitrary design choice of the nameplate power of the generator, without having to take into account historical wind data.  Thus, the structural cost of the generator is merely a linear function of the chosen design power specifications, not the imposed safety factor for a structure that must withstand all weather conditions.

Currently, our 100 sqm wing is equipped with a 16 mm diameter 3GPa ultimate tensile-strength line. The force may reach up to 600 kN before breaking[4].  When the wind is strong enough, the wing may exert forces certainly greater than the line's ultimate tensile strength.  The order of magnitude of such exceptional forces can raise the power of a single-wing high-altitude wind generator close to 900 MW; thus 30MN of traction, as also stated in the subject article preprint, confirming this finding. This is valid and generally correct only from a physical and geometrical point of view. Obviously, the technology cannot follow those requirements so closely. Another issue is the Capacity Factor of the machines which needs to be maximized for weak winds rather than preventively over-engineered for optimization in strong winds.

Thus, it is not practical to apply the features of wing/wind system interaction to the specifications of the generator.  It is better to find a compromise with reduced power that is more easily manageable and can be engineered/produced with a greater Capacity Factor because it requires less wind speed.

Following such design guidelines, it is the generator that shapes the dynamics of operation, controlling the lines through its system of pulleys. It is not necessary nor useful to have a full wing-speed-and-force profile dependent on the available natural resource. The arbitrarily chosen  3 MW nameplate generator will currently manage up to 300 kN of force (50% of the maximum load of the line ) when the wind speed does not exceed 15 m/s.  By regulating the operating altitude and wing flying position/orientation (pose) relative to the wind direction, sporadic stronger winds can be avoided and/or mitigated.

**Stronger and Constant High Altitude Wind Isn't the Original Enabling Factor**

Text quoted from page 2 paragraph 10:

*"[Developers] They currently rely on oversimplified approximations such as the logarithmic wind profile (Optiset al., 2016) or coarsely resolved reanalysis data sets (Archer and Caldeira, 2009) as the applicability of conventional spectral wind models (Burton, 2011) have not been verified for these altitudes."*

This statement is also not true and, again, potentially undermines the integrity of, and professional work accomplished, by KiteGen. We have already observed such an attitude in other publications where the authors try to gain some additional credibility, attempting to criticize the proceeding developments, their motive not clear [6].

At KiteGen, we prefer to be asked timely questions about our technical opinions or doubts about our assertions rather than be surprised after-the-fact. This is an open invitation to meet the team. In any case, it is a great opportunity to continue the discussion of this unprecedented opportunity.

KiteGen relies on two features/achievements regarding wind availability, which are precise and certainly not an oversimplification or approximation:

1)      Tethered airfoils can generate far more power than wind turbines simply because they can sweep a greater area for an equivalent or reduced expenditure of resources, since they would not incur the cost of the tower or be limited to the blade sizes that towers must accommodate. It is easy to compute such an increase in performance through Betz laws. In particular, the flying wings expose a lower Betz efficiency, compensated for by the larger area swept, which allows it to outperform the energy-harnessing potential of the wind turbine blades by a factor of three, assuming equal conditions of wind speed and aerodynamic surface.

2)      In 2003, KiteGen, in collaboration with Dutch astronaut Wubbo Ockels [1], gained insight into the potential of high, or tropospheric, winds; stronger and more constant than biosphere winds; then, in 2009, sought and obtained a study from an Italian research centre [5], providing

great amounts of data gained in Italy using Sodars, that made us fully aware of the greatly multiplied advantages at previously unforeseen and unexpected altitudes, creating a definitive and satisfactory solution to the wind issue.

Multiplying the threefold increase of performance due the "Betz" advantage, with an eightfold minimum increase of wind power at altitude, we obtain an astonishing exploitable natural resource that is at least *24 times* the typical wind turbine's harnessable power. This fact suggested that it was better to focus all efforts on the challenge of industrial-scale harnessing technology.

**Finally Some Positive Remarks Regarding the Technology**

Text quoted from page 16 paragraph 5:

*"Therefore, AWES need to be able to operate in a wide range of wind speeds or be controlled in a way that they avoid extreme conditions. The 12 months NoOBS simulation shows lower wind speeds than the 6 months simulations as the included summer months generally have lower wind speeds due to higher probability of unstable stratification. The Weibull fit of this simulation tends to overestimate higher wind speeds and underestimate low wind speeds at all altitudes."*

Text quoted from page 24 paragraph 10:

*"Using a simplified AWES model, assuming a constant tether length of 1500 m and neglecting drag and weight all data sets suggested an optimal operating altitude between 150 and 400 m. However, since stratification leads to a vast range of wind speed profiles AWES greatly benefit from dynamically adapting their operating altitude to maximize power production and minimize losses"*

Disregarding the fallacy of the over-simplification of the tropospheric wind generation model, those observations are certainly true, and finally desirable and quite easy to obtain. The wing's directional freedom effectively deals with extreme conditions and provides an effective adaptation opportunity. Stronger winds will not be exploited by regulating the operative altitude and/or the wing flying position compared to the wind direction (exiting the power spot by not flying crosswind).  That being said, there are a lot of automatic controls and engineering solutions to ensure safety and to manage modulation, transient conditions and all the possible issues that may arise when dealing with this natural resource.

**Optimal Operating Altitude and Power Production: Is It Really Required?**

Text quoted from page 22 paragraph 5:

*"Figure 13 summarizes the probability distribution of optimal operating altitude and optimal power with the white solid line showing the cumulative frequency of optimal operating altitude. Both simulations for this particular location and time period show similar trends with the most probable optimal altitude between approximately 200 and 400 m. Times of very high traction power are fairly rare and likely associated with low level jets. Lower power at higher altitudes is caused by the misalignment losses. Here we assume a constant tether length of 1500 m."*

This article is addressing a wind resource assessment totally outside of the KiteGen dynamic, unfortunately risking to be useless to both developers and researchers in general, which again falls into the error of inappropriate exaggeration. Although theoretically correct, the proposed wind environments for deployment are technologically unreachable. In fact, our 100 m2 wing, according to the authors, would reach a peak of 900MW. In short, a single wing would generate power almost equivalent to that from a *nuclear power plant*!

The out-of-scale wing 8-9MW/m$^2$ must be compared with the desired trend in wind turbine to reduce the specific power addressed, and the requirement of our giant wing that is even lower: 30kW/m$^2$ of wing surface or 300W/m$^2$ of specific power of the wind front

[Figure]

Subject article figure 13 here modified to highlight the scale mismatching

The plausibility of the project needs to preserved by correcting such inaccuracies, arguing the uselessness of this approach and introduce and make it clear that the most valuable feature of KiteGen is its baseload behavior, which is achieved without forcing it into optimization. It is better to avoid exploiting intense wind and address development toward obtaining nominal power with weak or very weak winds. Losing wind power when it is excessive, on the other hand, can be quite easily addressed, even though this was the biggest obstacle during the operation of our research prototypes, regularly leading to damaging some components of the equipment.

[Figure]

The KiteGen Carousel is superior to the best baseload power plants, including coal, gas and nuclear. The data are coming from several available data and reanalyses. The pink area depicts wind speeds available in the temperate zones of the planet, actually better with respect to the global average. This means that the KiteGen Carousel needs a very low wind speed (about 7 m/s on average) to work at high capacity for more than 8300 hours per year, even at altitudes less than 3000 m, especially in energy intensive areas of the world (yellow balloon "A"). The pumping kites need more power density to work at a capacity factor greater than 6000 hours per year at even lower altitudes (about 10 m/s on average - yellow balloon "B"). Wind turbines work

for 2000 equivalent hours per year, requiring a minimum wind speed of 12-14 m/s or 1100-1646 W/m2 to provide nominal power and is also reported the recent announced prototype of GE Haliade-X (yellow balloon "D") that is claimed to have a potential capacity factor of 63% or 5500h in very lucky sites. The Haliade-X abnormal data is in fact a merely and different commercial strategy, if compared to competitors, the turbine is geometrically and economically a 36 MW wind machine with the nameplate derated to 12 MW, this adjust the denominator of the CF formula (MWh/MW), unfortunately the expected manufacturing batch cost per unit is close to 150M€ while the prototype including the industrial tooling was announced at 400M€, skyrocketing the LCOE of the envisageable batch produced units to over €300/MWh, two order of magnitude higher compared to the LCOE of the "dematerialised" KiteGen Carousel.

The extreme optimization issue is a common thread to practically all the (pseudo) scientific articles generated by third parties which ignores grid requirements and energy quality, then is easily ridiculed, consequently leading to the potentially damaging underestimation of the value of the project, thus sharing the same errant promotion of photovoltaic and conventional wind turbines, when KiteGen is obviously the long-awaited solution to such issues.

**References**

[1] Ippolito, M. Ockels, W. (2003)."Kite Wind Generator, smart control of power kites for renewable energy production" submitted in PRIORITY 6.1 "Sustainable Energy Systems" Call FP6-2003 -TREN-2

Retrieved from http://energykitesystems.net/KiteGen/2003meeting56pages.pdf

[2] Cherubini, A., Papini, A., Vertechy, R., and Fontana, M. (2015). Airborne Wind Energy Systems: A review of the technologies, *Renewable and Sustainable Energy Reviews*, 51, 1461–1476, doi:10.1016/j.rser.2015.07.053,

Retrieved from http://linkinghub.elsevier.com/retrieve/pii/S1364032115007005

[3] Archer, C. L. and Caldeira, K. (2009). Global Assessment of High-Altitude Wind Power, *Energies*, 2, 307–319, doi:10.3390/en20200307,

Retrieved from http://www.mdpi.com/1996-1073/2/2/307/

[4] Ippolito, M Saraceno, E. (2019) KiteGen Research high altitude wind generation tropospheric wind exploitation under structural and technological constraints. *Research Gate* doi: 10.13140/RG.2.2.12701.97766

[5] Casale C., Silvano Viani S., Marcacci P. (2009) Valutazioni sui sistemi "kite wind generator" *CESI RICERCA* (in Italian) Retrieved from

https://docplayer.it/13336123-Valutazioni-sui-sistemi-kite-wind-generator.html

[6] Ippolito M. (2019) Reaction Paper to the Recent Ecorys Study KI0118188ENN.en.pdf Challenges for the commercialization of Airborne Wind Energy Systems

  DOI: 10.13140/RG.2.2.17236.24966 Retrieved from

https://www.researchgate.net/publication/331715736_Reaction_Paper_to_the_Recent_Ecorys_Study_KI0118188ENNenpdf_Challenges_for_the_commercialization_of_Airborne_Wind_Energy_Systems

---

## Short Comment (SC2) · 16 Apr 2019

Dear Authors,

congratulations for your very interesting and important analysis. I read the paper with great interest. I have one or two questions about the methods which lead to Fig. 3.

First, I am unsure about what you mean by "The continuous line in the left sub-figure represents the Root Mean Square Error (RMSE) of wind speed." This is just a question for clarification... The RMS of what? is it a measure of the temporal variation of the

measured or simulated wind speed on its own within some interval? Or an estimate of the precision of simulation or measurement (and if that, how is that uncertainty derived?)? Or an RMS of a difference between different quantities? I think the paper would profit if you could explain this in more detail.

The other question, which maybe is connected to the first question, is about the LI-DAR uncertainty. Your measurement itself is subject to an uncertainty, and it would be interesting if that would be clearly described.

Thank you very much! Philip Bechtle

---

## Editor Comment (EC1) · Jakob Mann (Editor) · 16 Apr 2019

The comment to the paper by Sommerfeld et al from Massimo Ippolito is not very specifically about the paper but rather a description of the virtues of a certain kite design. The comment in itself is not scientific in the sense that many of the claims are not carefully supported. Therefore I have asked the authors to disregard this comment and I will at the same time encourage M. Ippolito to submit a scientific paper to our journal on the subject that this comment is about. I will not remove the comment from the web because we want to keep a public track of all comments as long as they are

not outright personally offending or obvious nonsense.

Sincerely,

Jakob Mann, Editor-in-Chief.

---

## Referee Comment (RC1) · Rogier Floors (Referee) · 18 Apr 2019

**1   General comments**

The paper is a useful contribution to a better understanding of the winds at larger heights, which is not only relevant for the AWES applications for which the paper is written, but also in general for large wind turbines. Nudging with wind observations within the boundary layer has not been done a lot, so it is interesting to see how the

WRF model behaves. I have two major issues with the paper:

1) In the abstract it is stated that: "Observation nudging improves the overall accuracy of WRF". This cannot be concluded based on this study, because the observations are assimilated and then also used for evaluation. This will obviously result in the model being closer to the measurements, but this has nothing to do with WRF being more accurate 'overall'. If you want to draw this conclusion you would have to compare with measurements that are not assimilated in the model, preferably at some distance away from the point where the observations are nudged. Otherwise it should be more clearly written that the nudging is only valid at the lidar point: as it can be seen from Fig. 6 the modelled wind speed is just bias-corrected with approx. 1 m/s over a 180 km area, but it might well be that this detoriates wind speed comparison at other locations. For example, it could be that the bias at this point is caused by a wrong surface roughness or other local flow properties, which means the bias does not exists in other places. Also the nudging is likely only valid over land, because over sea the physical processes that determine the wind profile at a given time are different. All this should be written more clearly throughout the abstract/results/discussion/conclusion. Figures 2-6 all show the same message: nudging brings the model closer to the observations, so they can be combined into one or perhaps two figures. Figure 11 and 12 also show the same thing and can be combined.

2) The definition of the Obukhov length in Eq. 4 is not clear or wrong: to classify stability one should take into account the effect of the *virtual* kinematic sensible heat flux and not the dynamic sensible heat flux directly from WRF (W/m$^2$), which seems to be implied in Eq. 4 (although $H_s fc$ is not defined anywhere). In the WRF model surface layer fluxes are split up in a sensible and latent heat flux. Sensible and latent heat flux are equally important in a fairly moist areas as Germany (see for example Stull (2017) or Floors et al. (2013)), so they should both be used when computing the Obukhov length.

**2 Specific comments**

p3l8: It would be useful to give the opening angle of the lidar.

p4l2: What CNR threshold is used for filtering the data? What is the defintion of an 'available' measurement?

p4l6-9: I would remove this, because it has nothing to with the measurements, which is what the section is about. It is also discussing some of the results which have not yet been presented.

p4l13-17: All brackets make this section difficult to read. Please rewrite.

p4: Please mention the land-surface, radiation and surface-layer scheme that were used in the WRF model. p6l2: 180 km is a very large distance. See major comment 1.

p7l2: I assume the wind direction is not calculated like this because it would lead to discontinuities when crossing 360 degrees. Please add more details.

Section 4.1-4.3: see major comment 1;

p14l10-12: I think this is an important conclusion from this work and I agree that this is a potential application of using nudged WRF simulations. Perhaps it is useful to relate this to the discussion in Gryning et al. (2019) regarding the wind speed bias from lidars as a function of CNR threshold and data availability, to show that this issue is not specific for the site studied in your paper.

p16l9-11: The wind speed in summer is mostly lower due to the lower synoptic pressure gradients in that time of the year, not so much due to the stratification (particularly at greater heights).

p19 table 2: Maybe better to also express this as percentage instead of number of obs.

p19l7: It is not clear to me how the lidar measurements are normalized: with the friction velocity from the OBS run?

p26: Remove Appendix A, it is not discussed anywhere.

**3  Technical corrections**

p5l20: "(see equation: 2)" –> "(see Eq. 2)"

p9 Fig 4 label: Abbreviation HWS is not defined

p14l2: 100m –> 100 m (and m not in italics).

p17l2: to (Sommerfeld et al.) –> to Sommerfeld et al. Also I don't know the journal policy but usually you can only include references that are 'accepted' and not those that are 'in review'.

p17l4: ? –> ref

p20l3-4: These two lines repeat the same thing.

p20l5: ?? –> ref

p21l7: Please split equation and units.

p21l10: drag coefficient and drag coefficient? Also equal sign is not enclosed in '$'.

p22 Fig. 13 caption: there is mention of a),b),c) here but they are not in the figure.

p23l14: decreases –> decreases.

**4  References**

Floors, R., Vincent, C. L., Gryning, S.-E., Peña, A., Batchvarova, E. (2013). The Wind Profile in the Coastal Boundary Layer: Wind Lidar Measurements and Numerical Mod-

elling. Boundary-Layer Meteorol., 147(3), 469–491. http://doi.org/10.1007/s10546-012-9791-9

Gryning, S.-E., Floors, R. (2019). Carrier-to-Noise-Threshold Filtering on Off-Shore Wind Lidar Measurements. Sensors (Basel, Switzerland), 19(3). http://doi.org/10.3390/s19030592

Stull, R. (2017). Practical Meteorology: An Algebra-based Survey of Atmospheric Science. (Nina Horne, Ed.) (version 1.). Brooks/Cole

---

## Short Comment (SC3) · 19 Apr 2019

Although they don't apply specifically to the paper and the method proposed by the authors, in my opinion the remarks posted by M.Ippolito will be very useful to the authors in order to refine their findings and address some specific use case for their data, eventually in future works. It is true, indeed, that the players in HAWES (I find more correct the collective definition of High Wind Altitude Energy Systems, rather than Airborne Wind Energy that recalls the systems that fly a turbine) have different architectures that rely on different wind profiles at different altitudes. Try to determine a priori the best

wind and altitude conditions without distinguishing based on feasible architectures is less useful, from the point of view of the developer that reads the paper, than focusing on specific selected architectures, defining the best conditions for them. In other words maximizing peak power (as the focus on best traction conditions seems to suggest) or maximizing capacity factor (as in the architectural remarks from Ippolito) requires approaches that are very different from each other.

---

## Referee Comment (RC2) · Roland Schmehl (Referee) · 2 May 2019

**1   General comments**

This paper about an airborne wind energy resource assessment is a valuable contribution. The focus is clearly on the improvement of the wind speed forecast at higher altitudes using LiDAR data. A relatively small part is about the use of this wind data for the prediction of power production from AWES.

[Figure]

The description of the simplified power production model in Section 4.7 is unclear and inhomogeneous. On the one hand, very specific derivation steps of the original derivation are mentioned (geometric relation of aerodynamic force components and apparent wind velocity components) that are not of interest within the scope of this paper and would require proper illustrations and more background information. Other aspects that would be important are however not discussed, for example assumptions and specific choices. I recommend to carefully revise this part of the paper.

The original model of Schmehl et al (2013), that was also used as a basis for many other studies, is independent of tether length, as it is also apparent from your Equation (5). What was then the reason for you to choose a constant tether length of 1500 m? And how does the tether length come into play? This should be clearly described. If you would account for tether drag, the performance of the AWES would decrease with increasing tether length (compared to the idealized case of no tether drag). Tether drag could, for example, be taken into account by an additional drag contribution and lumping this to the kite, as some authors do. A possible reference could be van der Vlugt (2019). But I assume that this was not done in the paper, for the purpose of simplicity? If so, please state this, as it is important when considering large ranges of tether length.

For a implemented real AWES it makes generally sense to fly on a shorter tether when flying at lower altitudes, to reduce the effect of tether drag. For a pumping AWES, which is considered here, the tether length continuously varies. Assuming a constant tether length is seemingly in contradiction with this and should thus be motivated better. Just "Here we assume a constant tether length" is not sufficient in my opinion. I would also like to know, if the choice of the constant tether length could possibly influence the results displayed in Fig. 13 (for this is must be clarified how tether length actually enters the modeling).

**2 Specific comments**

**Authors**

I believe that the Fraunhofer IWES location at Bremerhaven, Germany, is meant, and not Oldenburg?

**Abstract**

I would spell out WRF once, as you do with AWES.

**Introduction**

Add a reference to Bechtle et al (2019). This could for example be done on p. 2, l. 14, just after Archer and Caldeira (2009).

Uwe Fechner (2016) describes in his dissertation and a later book chapter (https://doi.org/10.1007/978-981-10-1947-0_15) a turbulence model for AWES, based on the Mann turbulence model. As you shortly mention conventional spectral wind models (Burton, 2011) this might be worth a discussion point.

p. 2, l. 23: You state "No mid-altitude measurement device can reliably gather long term, high frequency data." but do not give any reason for this. This statement should also be better embedded in the surrounding text.

p. 2, l. 25: Your reference to future work (complementation of TI estimates with LES data) is better for the conclusions section.

**WESD**

p. 2, l. 28: Add a reference to the Onkites II project report, available from https://doi.org/10.2314/GBV:1009915452 Can the measurement data of OnKites II be made publicly available, as a data reference to complement this and the earlier paper? This would increase the value of this research tremendously (reproducibility!).

**Mesoscale Modeling Framework**

p. 4, l. 16: For the non-experts of this specific technique it would make sense to elaborate on the "non-physical forcing term". Why non-physical? Why not physical?

p. 4, l. 18: It is unclear what the use of 3 nested domains is. Please clarify. What is $\eta$-pressure? (also ""$\eta$-levels" in l. 23)

p. 4, l. 25: Again for the non-experts: what is the difference between "observation nudging" and "analysis nudging"? Maybe a pointer to the respective subsections, where you explain this, is sufficient.

p. 5, l. 4: What is the meaning of "$q_m$ interpolated"? And what means "$(q_0)$"?

p. 5, l. 9: "hydrostatic"? This paper is about atmospheric flows.

p. 5, l. 13: The time expression in the bracket is not correctly written. It is not the mathematical constant 2.71828 that is meant here, because this would lead to 9 seconds.

**Results**

Elaborate on how unavailability of LiDAR data is handled for the nudged simulations.

p. 8, l. 3: RSME is missing in legend.

p. 8, l. 5: The reduction of the spread of the bias is hard to observe by eye.

[Figure]

p. 8, l. 9: Doesn't nudging reduce the error? So, reduced nudging would result in larger error?

p. 9, l. 14: Please elaborate on this sentence.

p. 9, l. 11: Bechtle et al (2019) have used a similar representation as the one described here, using dots to show the optimal altitude for operation of an AWES. A reference should thus be added, and possibly also a discussion of the usefulness of this measure added (i.e. an AWES will generally sweep an altitude range, which means that this single point characterization is only a very rough measure.)

p. 9, l. 14: How do you see that the LLJ and the .... are weaker? I can hardly see anything.

p. 11, l. 7: You write "remain the same". Shouldn't $\Delta V$ be zero?

p. 11, l. 8: You write "change in wind speed": is this observed by the gradient?

p. 14-15, Figs. 7 and 8: Why are contour plots of Fig. 7 not as smooth as respective plots of Fig. 8. The caption mentions "filtered": aren't these "unfiltered"?

p. 20, l. 3: You write "We chose": how do you control this?

p. 21, l. 8: "Misalignment" is between TETHER and wind direction. It should be clear that the azimuth and elevation angles describe the angular position of the kite or aircraft with respect to the ground station. Renaming of $\theta$ as elevation angle is dangerous, because it is generally use for the polar angle.

**Conclusions**

I am missing some conclusions of Section 4.7 on the AWES power estimation.

[Figure]

**3 Language & style comments**

**General spelling**

- Use of dashes should be checked (e.g. "high-resolution data" or "long-term statistics" would be correct)

- Do not capitalize abbreviations (see https://www.aje.com/en/arc/editing-tip-capitalization-when-defining-abbreviations/).

**Title**

p.1: "Airborne Wind Energy" should be "Airborne wind energy".

**Abstract**

p. 1, l. 9: I would add an "it" between "but" and "becomes".

**Introduction**

p. 2, l. 4: "Airborne Wind Energy Systems" should be "Airborne wind energy systems".

p. 2, l. 4: I would say that AWES are a class of renewable energy technologies, and not a source of energy. The source is the wind.

p. 2, l. 10: Instead of "marketplace" I would just write "market".
p. 2, l. 11: ...none are YET commercially available.

p. 2, l. 13: "power" should be "power output" as you list wind energy technologies here.

p. 2, l. 23: ... variations to resolved quantities are parametrized. The "resolved" sounds wrong and the meaning of this sentence is also not clear to me.

p. 2, l. 25: "here presented" -> "presented in this study"

p. 2, l. 27: Year is missing in reference.

p. 3, l. 1: "power" -> "power output".

**Measurement campaign**

p. 4, l. 2: Year is missing in reference.

p. 4, l. 3: Should be "emphasizes".

p. 4, l. 6: Should be "... the WRF-calculated...". The entire expression "WRF-calculated sensible surface heat flux (SHF)" sounds incomprehensible to me. What is the role of the "sensible"?

p. 4, l. 9: Should be "... the SHF".

I would move Fig. 1 to the next section and remove the reference to the white X here. Because in the next section you explain the 3 hierarchically nested domains used for the WRF. Here, in this section, the figure introduces more questions than answers.

**Mesoscale modeling Framework**

p. 4, l. 14: Year is missing in reference. It is also not clear whether the "section 2" in the referenced paper or the present one is meant.

p. 4, l. 15: Year is missing in reference.

p. 4, l. 17: Why discussing here spatial resolutions when this is all given in Table 1?

p. 4, l. 23: "Turbulent Kinetic Energy" should be "Turbulent kinetic energy". Add "(TKE)" here and use the abbreviation in the next sentence.

p. 4, l. 29: Maybe a footnote link with the URL is better? This bibliographic reference looks strange.

p. 5, l. 20: Something is wrong after $W_x y$.

**Results**

p. 6, l. 5: Should be "differences".

p. 7, Fig. 2, legends: text and number should be separated by a space and also a comma.

p. 9, l. 2: Replace "bias" by "error".

p. 17, l. 3: Reference missing (?).

p. 19, Table 2: last three columns in % of time would be better readable.

p. 20, l. 4: "additional two" -> "two additional"

p. 20, l. 5: Figure reference is missing (??).

p. 21, l. 6: Why do you use a subscript "air" for the density? This study is only about

atmospheric flows, so the index can be safely omitted.

p. 21, l. 7: Set equation in displaymode.

p. 21, l. 10: "... are assumed constant are ...": something is wrong here.

**Conclusions**

p. 23, l. 7: Six months OF LiDAR .....

p. 23, l. 13: Dot behind "decreases" is missing.

**Appendices**

p. 25, Figure A1: what means the question mark at the end of this caption? "U profile" -> "velocity profile".

p. 25-26, Table A1: Include in the caption to which software & version, possibly also model, these settings refer.

**References**

There are many references for which the DOI is occurring twice, as "doi:..." and as URL "https://doi.org/...".

p. 28, l. 13: what is this oCLC number? I would use either ISBN or DOI.

p. 28, l. 19: Publisher or standard-issuing organization is missing.

p. 29, l. 15: Correct the URL.

p. 29, l. 23: Add DOI.

p. 29, l. 27: Insert comma/dot and space between "2016" and "At".

p. 30, l. 3: "Statistik" should be starting with a capital "S", according to German spelling. Consider choosing an English textbook as reference.

p. 30, l. 8: This is a contributed chapter in a book. Accordingly the reference should be Schmehl, R., Noom, M., van der Vlugt, R.: Traction Power Generation with Tethered Wings. In: Ahrens, U., Diehl, M., Schmehl, R. (eds.) Airborne Wind Energy. Springer, Berlin Heidelberg, 2013. doi:10.1007/978-3-642-39965-7_2

p. 30, l. 14: Update this.

**4  References**

van der Vlugt, R., Bley, A., Noom, M., Schmehl, R.: Quasi-Steady Model of a Pumping Kite Power System". Renewable Energy, Vol. 131, pp. 83-99, 2019. doi:10.1016/j.renene.2018.07.023

Bechtle, P., Schelbergen, M., Schmehl, R., Zillmann, U., Watson, S.J.: Airborne wind energy resource analysis. Renewable Energy, Vol. 141, pp. 1103-1116, 2019. doi:10.1016/j.renene.2019.03.118

Mann, J.: The spatial structure of neutral atmospheric surface-layer turbulence. Journal of Fluid Mechanics273, 141 (1994). doi:10.1017/S002211209400188620

Mann, J.: Wind field simulation. Probabilistic Engineering Mechanics13(4), 269–282 (1998). doi:10.1016/S0266-8920(97)00036-2

---

## Author Comment (AC1) · 29 Jun 2019

Dear Dr. Floors,

Thank you very much for your helpful review of our manuscript, "Improving mid-altitude mesoscale wind speed forecasts using LiDAR-based observation nudging for Airborne Wind Energy Systems", wes-2019-7. We have modified the manuscript accordingly, removed or consolidated several figures and adapted the text.

Please find our response to individual comments below.

Changes are highlighted in the "Supplementary Material" pdf. Text and figures marked in red were removed from the original submission and replaced by text and figures marked in blue. Following are our replies to your comments and a description of modification to the manuscript.

Sincerely,
Markus Sommerfeld

**Comments by the Referee**

0.1 General comments

*The paper is a useful contribution to a better understanding of the winds at larger heights, which is not only relevant for the AWES applications for which the paper is written, but also in general for large wind turbines. Nudging with wind observations within the boundary layer has not been done a lot, so it is interesting to see how the WRF model behaves. I have two major issues with the paper:*

1. *In the abstract it is stated that: "Observation nudging improves the overall accuracy of WRF". This cannot be concluded based on this study, because the observations are assimilated and then also used for evaluation. This will obviously result in the model being closer to the measurements, but this has nothing to do with WRF being more accurate 'overall'. If you want to draw this conclusion you would have to compare with measurements that are not assimilated in the model, preferably at some distance away from the point where the observations are nudged. Otherwise it should be more clearly written that the nudging is only valid at the lidar point: as it can be seen from Fig. 6 the modelled wind speed is just bias-corrected with approx. 1 m/s over a 180 km area, but it might well be that this detoriates wind speed comparison at other locations. For example, it could be that the bias at this point is caused by a wrong surface roughness or other local flow properties, which means the bias does not exists in other places. Also the nudging is likely only valid over land, because over sea the physical processes that determine the wind profile at a given time are different. All this should be written more clearly throughout the abstract/results/discussion/conclusion. Figures 2-6 all show the same message: nudging brings the model closer to the observations, so they can be combined into one or perhaps two figures. Figure 11 and 12 also show the same thing and can be combined.*
- The sentence in the abstract has been changed to a more precise formulation: 'Observation nudging improves the WRF accuracy at the measurement location.'

- Wind conditions and WRF simulation offshore are not subject of this manuscript and no investigation of observation nudging offshore have been performed. While we assume that the non-physical nature of observation nudging impacts the flow offshore as well, we can not draw this conclusion at. We therefore would prefer not mentioning offshore conditions in this paper.

- While figure 2-6 all show the impact of observation nudging, they present different aspects. Figure 2 visualizes the correlation between measurements and absolute occurrence of certain wind conditions, figure 3 shows the overall altitude dependent impact of nudging, figure 4 shows a representative day and figure 5 and 6 shows the spatial impact of observation nudging. We see that figure 5 and 6 show similar results and chose to remove figure 6.

- Figure 11 shows the mean wind speed profiles categorized based on Obukhov length whereas figure 12 shows the purely mathematical k-means clustered wind speed profile probability distribution. This mathematical approach was chosen due to the lack of heat flux and temperature measurements and to be consistent with "LiDAR-based characterization of mid-altitude wind conditions for airborne wind energy systems" doi: 10.1002/we.2343. Since the WRF simulations provide all this information we decided to represent the Obukhov length categorized data in the same way as the k-means clustered data previously.

2. *The definition of the Obukhov length in Eq. 4 is not clear or wrong: to classify stability one should take into account the effect of the \*virtual\* kinematic sensible heat flux and not the dynamic sensible heat flux directly from WRF (W/m2), which seems to be implied in Eq. 4 (although Hsfc is not defined anywhere). In the WRF model surface layer fluxes are split up in a sensible and latent heat flux. Sensible*

*and latent heat flux are equally important in a fairly moist areas as Germany (see for example Stull (2017) or Floors et al. (2013)), so they should both be used when computing the Obukhov length.*

- We adjusted our calculation of the Obukhov length in equation 4. The equation, which was taken from Sempreviva and Gryning, 1996 "Humidity fluctuations in the marine boundary layer measured at a coastal site with an infrared humidity sensor", now takes latent and sensible surface heat flux into account.
- $OL = \left( \frac{-u_*^3 \theta_v}{kg} \right) \left( \frac{1}{Q_S} + \frac{0.61}{Q_L \theta} \right)$
- Equation 18.16 just uses the kinematic surface heat flux in Stull,2017: Practical Meteorology An Algebra-based Survey of Atmospheric Science

**0.2 Specific comments**

- *p3l8: It would be useful to give the opening angle of the lidar.*

  – Added the opening angle in brackets: 62 degree or 28 degree to the horizon.

- *p4l2: What CNR threshold is used for filtering the data? What is the definition of an 'available' measurement?*

  – We used a self-defined filter described in the "LiDAR-based characterization of mid-altitude wind conditions for airborne wind energy systems" doi: 10.1002/we.2343. We added a short description in this manuscript and refer to the previously mentioned paper for detailed information. Data availability is defined as the time when useful data (not filtered out) is available divided by the total time of the measurement period (6 months).

- *p4l6-9: I would remove this, because it has nothing to with the measurements, which is what the section is about. It is also discussing some of the results which have not yet been presented.*

  – Agreed and removed.

- *p4l13-17: All brackets make this section difficult to read. Please rewrite.*

  – Removing these brackets is difficult as some of them are due to the bibliography style, reference to figures and the definition of new abbreviations.Rewrote the definition of NoOBS and OBS in a sub-clause and removed the brackets around the WRF version.

- *p4: Please mention the land-surface, radiation and surface-layer scheme that were used in the WRF model.*

  – land-surface option: sf_surface_physics: 4, Noah-MP land-surface model (see additional &noah_mp namelist)
  – long wave: ra_lw_physics: 1, rrtm scheme
  – short wave: ra_sw_physics: 1, Dudhia scheme
  – radt: 18,6,2 min between physics calls
  – surface-layer option: sf_sfclay_physics: 5, MYNN surface layer
  – reference: https://esrl.noaa.gov/gsd/wrfportal/namelist_input_options.html

- *p6l2: 180 km is a very large distance. See major comment 1.*

  – The radius of 180 km is chosen so that the entire inner domain is affected by obs nudging and the spatial impact can be quantified.
  – changed subsentence to:"... thereby affecting the whole inner domain."

- *p7l2: I assume the wind direction is not calculated like this because it would lead to discontinuities when crossing 360 degrees. Please add more details.*

  – Angular difference is calculated by using *angdiff* in Matlab. The results are wrapped on the interval $[-\pi, \pi]$. Added a sub-clause to this point.

- *Section 4.1-4.3: see major comment 1;*

  – See response to comment 1.

- *p14l10-12: I think this is an important conclusion from this work and I agree that this is a potential application of using nudged WRF simulations. Perhaps it is useful to relate this to the discussion in Gryning et al. (2019) regarding the wind speed bias from lidars as a function of CNR threshold and data availability, to show that this issue is not specific for the site studied in your paper.*

  – Added this reference. We agree that it is good to point out that this is not a site specific issue.
  – Added to conclusion:
    * The bias between real and LiDAR measured wind speed, which depends on the applied CNR threshold and data availability, can result in a misrepresentation of the actual wind conditions especially at higher altitudes. Mesoscale models, particularly with observation nudging, can be used to account for this error.

- *p16l9-11: The wind speed in summer is mostly lower due to the lower synoptic pressure gradients in that time of the year, not so much due to the stratification (particularly at greater heights).*

  – Changed this sentence to reflect this fact.

- *p19 table 2: Maybe better to also express this as percentage instead of number of obs.*

- – Adapted the table.

- *p19l7: It is not clear to me how the lidar measurements are normalized: with the friction velocity from the OBS run?*

  - – Clarified formulation. Simulated friction velocity and heat flux is used to categorize and normalize LiDAR data.

- *p26: Remove Appendix A, it is not discussed anywhere.*

  - – removed figures in appendix A1.

**0.3  Technical corrections**

- *p5l20: "(see equation: 2)" –> "(see Eq. 2)"*

- *p9 Fig 4 label: Abbreviation HWS is not defined*

- *p14l2: 100m –> 100 m (and m not in italics).*

- *p17l2: to (Sommerfeld et al.)  –> to Sommerfeld et al.  Also I don't know the journal policy but usually you can only include references that are 'accepted' and not those that are 'in review'.*

- *p17l4: ? –> ref*

- *p20l3-4: These two lines repeat the same thing.*

- *p20l5: ?? –> ref*

- *p21l7: Please split equation and units.*

- *p21l10: drag coefficient and drag coefficient? Also equal sign is not enclosed in '$'.*

- *p22 Fig. 13 caption: there is mention of a),b),c) here but they are not in the figure.*

- *p23l14: decreases –> decreases.*

- All technical corrections above have been addressed.

Please also note the supplement to this comment:
https://www.wind-energ-sci-discuss.net/wes-2019-7/wes-2019-7-AC1-supplement.pdf

**Supplement:**

[revised manuscript text omitted]

~~Unstable and stable stratifications were identified by partitioning the data based on the sign of WRF-calculated sensible surface heat flux (SHF). These two atmospheric conditions lead to bi-modal wind speed probability distributions aloft which is not adequately represented by a single Weibull distribution fit. Mid-altitude wind speeds are better represented by the weighted sum of two wind speed probability density function (PDF) fits conditioned by the sign of the SHF.~~

[Figure]

(a) Three nested WRF domains                                    (b) Inner WRF domain

**Figure 1.**

**3  Mesoscale Modeling Framework**

To complement the 6 months LiDAR data set two WRF  simulations using the advanced research weather research and forecasting (ARW) model (Skamarock and Klemp, 2008) were carried out. The 'baseline run' , which is hereinafter referred to as *NoOBS*,  is a 12 month study of the area around the measurement location (see figure 2) from the 1st of September 2015 used to derive annual statistics. LiDAR measurements (Sommerfeld et al., 2019)  were incorporated into the six

5    months test model between September 2015 and February 2016 using *OBSGRID* (Wang et al., 2015) , which is hereinafter referred to as *OBS*.

This methodology uses the difference between model and measurements to calculate a non-physical forcing term  which is added to the governing conservation equations of the simulation to gradually nudge the model towards the observation (see equation 1) (Stauffer et al., 1991; Deng et al., 2007). Each simulation is composed of three nested domains with 27-, 9-

10   and 3-km grid spacing and horizontal grid dimensions of about $120 \times 120$ elements at 60  heights along the terrain following vertical hybrid pressure coordinate $\eta$. Differences between the simulation runs (see section 3.1) are compared within the innermost domain of the simulation. Output data was stored in 10 min intervals. Figure 2 shows the topography map of the simulation. Initial and boundary conditions of both simulations are based on the *ERA-Interim* (Dee et al., 2011) reanalysis data set by the European centre for medium-range weather forecasts (ECMWF) which consists of 6 hourly atmospheric fields with a

15   spatial resolution of roughly 80 km horizontally and 60 $\eta$ levels. Turbulent Kinetic Energy closure within the ABL was achieved

[Figure]

(a) Three nested WRF domains          (b) Inner WRF domain

**Figure 2.** Topography map of all three WRF model domains (a) and a magnification of the innermost domain (b) with the LiDAR measurement site marked by a white X.

by using the Mellor Yamada Nakanishi Niino (MYNN) 2.5 scheme which predicts sub-grid turbulent kinetic energy (TKE) as a prognostic variable (Nakanishi and Niino, 2004; Lee and Lundquist, 2017). The Noah-MP land-surface model, MYNN surface layer scheme were used. The rrtm long wave radiation and Dudhia short wave radiation scheme were used (see: table A1 in the appendix). In addition to observation nudging (see subsection 3.1) analysis nudging was performed on every domain of each simulation .  Analysis nudging nudges each grid point  towards a time-interpolated value from gridded

5  analyses of synoptic observations (Stauffer et al., 1991) whereas observation nudging directly drives the simulation towards the additional observations. Within the planetary boundary layer (PBL) of the inner domain analysis nudging was switched off (see nudging settings in table A1 in the appendix). All simulations were run on the *EDDY*  [1] High-Performance Computing clusters at the University of Oldenburg.
* * *
[1]EDDY: HPC cluster at the Carl von Ossietzky Universität Oldenburg, see: https://www.uni-oldenburg.de/fk5/wr/hochleistungsrechnen/hpc-facilities/eddy/

**Table 1.**

[revised manuscript text omitted]

~~The effect of observation nudging on horizontal wind speed remains almost unchanged along lines of constant longitude or latitude. The difference peaks between 400 m and 600 m and drops towards higher altitudes (as seen in figure 7) which shows the average absolute difference in wind speed along a slice of constant longitude and latitude through the center cell of the inner domain for the entire simulation period. Wind speeds at low and high-altitudes are less affected by nudging while OBS~~

15

(a) Slice along constant longitude

(b) Slice along constant latitude

**Figure 7.**

[revised manuscript text omitted]

In comparison with the unnudged simulation, OBS shows an increase in unstable and near unstable situations. Stable and near stable stratification seems almost unaffected by OBS nudging, while neutral and very stable stratification occur slightly less often. This might improve the overall predicting capabilities of WRF as the MYNN 2.5 boundary layer scheme overestimates the frequency of very stable conditions with an error of up to 9 % (Krogsæter and Reuder, 2015). Neutral conditions, still
5   commonly used in many wind energy siting applications, only occur about 30 % of the time during the measurement period and only about 20 % of the time during the one year reference NoOBS simulation.

Figure 12 shows the mean wind speed profiles categorized and normalized by the corresponding friction velocity $u_*$. We assumed that The nudged simulation OBS is assumed to be sufficiently close to the measurements and is therefore used to normalize and the same categorization as in OBS, since no measurements were available to determine $L_{LiDAR}$. All profiles follow expected trends with unstable profiles showing the smallest wind shear and stable profiles showing the largest. Altitudes below 200 m are least affected by observation nudging as OBS remains almost unchanged from NoOBS (see section 4.1). Both models are in good accordance with measurements during unstable and near unstable conditions. The stable and very stable profiles of the unnudged simulation show a peak at around 300 m which is indicative of a characteristic low level jet. The more irregular trend of the very stable LiDAR data set could be caused by a small sample size since only 5 % of the overall data is considered very stable.

[Figure]

**Figure 12.** 6 months mean HWS profiles of LiDAR, OBS and NoOBS data classified by stability class defined by Obukhov length (table 3).

Expanding on the previous approach (subsection 4.5) of splitting the data into times of positive and negative SHF to differentiate states of unstable and stable stratification, we make use of k-means clustering (Lloyd, 1982) to identify two additional sub-states : stable and very stable as well as unstable and shear driven. to better differentiate between the different flow situations. We chose To differentiate additional two sub-states which identify stable and very stable as well as unstable and shear driven conditions. LiDAR results for reference can be found in (Sommerfeld et al., 2019).

Figures 14 to ?? shows the probability distribution of each sub-state the different stability categories for each model simulation with the cluster centroids mean highlighted by white squares. All clusters categorize show distinct trends and distributions that are consistent between data sets , which contribute to the multi-modality of the overall wind speed probability

distribution. The difference in high-altitude wind speeds between stratifications indicate the influence of different geostrophic wind conditions. The categorization by $OL$ is based on surface data and seems to be valid within the lower part of the atmosphere where the spread of the corresponding probability distribution is relatively small in comparison to high altitudes. This is particularly true for stable and neutral stratification where wind speeds above approximately 200 m spread widely. Unstable conditions are probably more consistent because of increased mixing up to high altitudes. Altitudes below 200 m

5 are least affected by observation nudging as OBS remains almost unchanged from NoOBS (see section 4.1). Stable profiles show a peak at around 300 m which is indicative of a characteristic low level jet. Comparing OBS and NoOBS 6 months, observation nudging seems to reduce the spread at higher altitudes within each category except very stable. The impact of observation nudging on wind profiles during unstable stratification is relatively low while wind speed profiles under neutral and stable stratification are more affected.

10 ~~stratification with low wind shear all the way up to 1100 m. The top right plot shows statistics of shear-driven wind profiles that occur during times of positive SHF. Stable (bottom right) and very stable stratification (bottom left) are characterized by strong wind shear and higher average wind speeds. NoOBS predicts a higher chance of wind speed reduction during very stable stratification above 600 m while wind speeds in OBS steadily increase up to 1100 m. Both models indicate the existence of LLJ during stable stratification between 200 and 400 m. The difference in high-altitude wind speeds indicate the influence of~~

15

[Figure]

**Figure 13.**

[revised manuscript text omitted]

---

## Author Comment (AC2) · 29 Jun 2019

Dear Prof. Schmehl,

Thank you very much for your helpful review of our manuscript, "Improving mid-altitude mesoscale wind speed forecasts using LiDAR-based observation nudging for Airborne Wind Energy Systems", wes-2019-7. We have modified the manuscript accordingly, removed or consolidated several figures and adapted the text.

Please find our response to individual comments below.

Changes are highlighted in the "Supplementary Material" pdf. Text and figures marked in red were removed from the original submission and replaced by text and figures marked in blue. Following are our replies to your comments and a description of modification to the manuscript.

Sincerely,
Markus Sommerfeld

[Figure]

**Comments by the Referee**

**0.1 General comments**

*This paper about an airborne wind energy resource assessment is a valuable contribution. The focus is clearly on the improvement of the wind speed forecast at higher altitudes using LiDAR data. A relatively small part is about the use of this wind data for the prediction of power production from AWES.*

*The description of the simplified power production model in Section 4.7 is unclear and inhomogeneous. On the one hand, very specific derivation steps of the original derivation are mentioned (geometric relation of aerodynamic force components and apparent wind velocity components) that are not of interest within the scope of this paper and would require proper illustrations and more background information. Other aspects that would be important are however not discussed, for example assumptions and specific choices. I recommend to carefully revise this part of the paper.*

- The sentence: "Additional losses caused by gravity, tether sagging and tether drag are neglected" summarizes some of the assumptions
- steady state assumption, constant $c_L$ and $c_D$ are mentioned in the text
- added tether sagging and point-mass assumption
- removed equations and simplified the derivation.

*The original model of Schmehl et al (2013), that was also used as a basis for many other studies, is independent of tether length, as it is also apparent from your Equation (5). What was then the reason for you to choose a constant tether length of 1500 m? And how does the tether length come into play? This should be clearly described.*

- Agreed.

- The equation is a function of elevation angle. The optimal elevation angle is calculated from the constant tether length and operating altitude.

- Added additional sentences:

  - Optimal elevation angle ($\varepsilon_{opt}$) and operating altitude ($z_{opt}$) are geometrically related to the assumed to be constant tether length ($L_{tether}$) of 1500 m ($\sin \varepsilon_{opt} = \frac{z_{opt}}{L_{tether}}$).
  - The tether length of each estimation is assumed to be constant and used to calculate the optimal elevation angle.

- Do you think the equation $\sin \varepsilon = \frac{altitude}{L}$ is necessary?

*If you would account for tether drag, the performance of the AWES would decrease with increasing tether length (compared to the idealized case of no tether drag). Tether drag could, for example, be taken into account by an additional drag contribution and lumping this to the kite, as some authors do. A possible reference could be van der Vlugt (2019). But I assume that this was not done in the paper, for the purpose of simplicity? If so, please state this, as it is important when considering large ranges of tether length. For a implemented real AWES it makes generally sense to fly on a shorter tether when flying at lower altitudes, to reduce the effect of tether drag.*

- Yes this is out of scope for this paper.

- The following sentences already address the additional losses associated with a longer tether:

  - All estimates show diminishing benefits of a longer tether. These incremental gains would probably be negated by additional drag and weight associated losses.

*For a pumping AWES, which is considered here, the tether length continuously varies. Assuming a constant tether length is seemingly in contradiction with this and should thus be motivated better. Just "Here we assume a constant tether length" is not sufficient in my opinion. I would also like to know, if the choice of the constant tether length could possibly influence the results displayed in Fig. 13 (for this is must be clarified how tether length actually enters the modeling).*

- See previous comments above

**0.2 Specific comments**

**0.2.1 Authors**

- *I believe that the Fraunhofer IWES location at Bremerhaven, Germany, is meant, and not Oldenburg?*

    – While the headquarter of Fraunhofer IWES is in Bremerhaven, IWES has several other locations in Oldenburg, Bremen, Hannover, Bochum and Hamburg. Gerald Steinfeld and Martin Dörenkämper work from Oldenburg.

**0.2.2 Abstract**

- *I would spell out WRF once, as you do with AWES.*

    – A definition of WRF was added to the abstract.

**0.2.3 Introduction**

- *Add a reference to Bechtle et al (2019). This could for example be done on p. 2, l. 14, just after Archer and Caldeira (2009).*

– Reference added.

- *Uwe Fechner (2016) describes in his dissertation and a later book chapter a turbulence model for AWES, based on the Mann turbulence model. As you shortly mention conventional spectral wind models (Burton, 2011) this might be worth a discussion point. (https:// doi.org/ 10.1007/ 978-981-10-1947-0_15)*

    – Reference added.

- *p. 2, l. 23: You state "No mid-altitude measurement device can reliably gather long term, high frequency data." but do not give any reason for this. This statement should also be better embedded in the surrounding text.*

    – Added a sentence and references

- *p. 2, l. 25: Your reference to future work (complementation of TI estimates with LES data) is better for the conclusions section.*

    – Sentence removed and added to conclusion section.

- *p. 2, l. 28: Add a reference to the Onkites II project report, available from https://doi.org/10.2314/GBV:1009915452 Can the measurement data of OnKites II be made publicly available, as a data reference to complement this and the earlier paper? This would increase the value of this research tremendously (reproducibility!).*

    – Reference added. I can not make the decision to publish the data myself and have to refer you to Adrian Gambier and Julia Gottschall.

**0.2.4 Mesoscale Modeling Framework**

- *p. 4, l. 16: For the non-experts of this specific technique it would make sense to elaborate on the "non-physical forcing term". Why non-physical? Why not*

*physical?*

  – The additional term is added to the conservation equations that guide the simulation. It is non-physical in nature since it is not based on any physical principle in contrast to the conservation of mass and momentum for example from which the conservation equations are derived. This additional term which is driven by the difference between measurement and simulation nudges the simulation closer to measurement without creating discontinuities in the simulation.

  – Added a subordinate clause: "...non-physical forcing term which is added to the governing conservation equations of the simulation to gradually nudge..."

• *p. 4, l. 18: It is unclear what the use of 3 nested domains is. Please clarify. What is $\eta$-pressure? (also "$\eta$-levels" in l. 23)*

  – The sentence explains the benefit of nested domains which is that the inner domains have higher spatial and temporal resolution.

  – added: ... along the terrain following vertical hybrid pressure coordinate $\eta$.

• *p. 4, l. 25: Again for the non-experts: what is the difference between "observation nudging" and "analysis nudging"? Maybe a pointer to the respective subsections, where you explain this, is sufficient.*

  – Added an additional sentence explaining the difference between analysis and observation nudging:

    ∗ For analysis nudging each grid point is nudged towards a time-interpolated value from gridded analyses of synoptic observations whereas observation nudging directly drives the simulation towards observations.

  – Reference: In analysis nudging, the model fields are nudged at every grid point toward an analysis of the observations on the model grid in a manner such that the nudging term is proportional to the difference between the model
and the analysis at each grid point (ref: https://pdfs.semanticscholar.org/1c94/a18e5ce2edd5fa5189dc293d8d33fe46b7c7.pdf)

- *p. 5, l. 4: What is the meaning of "qm interpolated"? And what means "(q0)"?*
  - This sentence explains equation 1 which defines the additional forcing term introduces by observation. This forcing term is driven by the difference between observation $q_o$ and model $q_m$
  - $q_m$ is the modeled quantity (e.g. wind velocity component, temperature, humidity etc...) which has to interpolated to the location of the additional observation due to the large grid size of mesoscale simulations
  - $q_o$ is the observed quantity
  - No changes to the manuscript

- *p. 5, l. 9: "hydrostatic"? This paper is about atmospheric flows.*

  - Correct. This is part of the model.

- *p. 5, l. 13: The time expression in the bracket is not correctly written. It is not the mathematical constant 2.71828 that is meant here, because this would lead to 9 seconds.*

  - replaced by: $1/6 \ 10^{-4}s$

**0.2.5 Results**

- *Elaborate on how unavailability of LiDAR data is handled for the nudged simulations.*

  - Added sentence in subsection 'Observation Nudging': Nudging could not be performed at times and altitudes where LiDAR data was not available.

- *p. 8, l. 3: RSME is missing in legend.*

  – added RMSE to the legend

- *p. 8, l. 5: The reduction of the spread of the bias is hard to observe by eye*

  – The reduction in bias spread (visualized by the lengths of the horizontal blue and red lines, also called whiskers) is clearly visible in the PDF.

- *p. 8, l. 9: Doesn't nudging reduce the error? So, reduced nudging would result in larger error?*

  – Good point. Subordinate clause removed.

- *p. 9, l. 14: Please elaborate on this sentence.*

  – Need more information. page 9 is referenced n this and the following comments. Probably page 8

  – added: ... as can be seen in the right box plot in figure 3

- *p. 9, l. 11: Bechtle et al (2019) have used a similar representation as the one described here, using dots to show the optimal altitude for operation of an AWES. A reference should thus be added, and possibly also a discussion of the usefulness of this measure added (i.e. an AWES will generally sweep an altitude range, which means that this single point characterization is only a very rough measure.)*

  – While we agree that a discussion is useful, we disagree that the usage of dots to visualize optimal altitude justifies the added reference. The same reference as been cited earlier in the paper.

  – Added sentence: A single point is only a rough measure of operational altitude since AWES generally sweep a range of altitudes.

- *p. 9, l. 14: How do you see that the LLJ and the .... are weaker? I can hardly see anything.*

  – The color of the contour plot in the upper subplot, which is refers to the horizontal wind speed, is significantly different between OBS and NoOBS.

- *p. 11, l. 7: You write "remain the same". Shouldn't $\Delta V$ be zero?*

  – That is right. While the boundary condition ($\Delta U = 0$) is applied on the outward facing surface of the cubic grid cell the wind speed values are stored and interpolated to the center of the grid cell. This leads to $\Delta U \neq 0$ at the boundary grid cell.

  – Added sentence: $\Delta U \neq 0$ because wind speed values are interpolated to the center of each grid cell.

- *p. 11, l. 8: You write "change in wind speed": is this observed by the gradient?*

  – can be seen by the spike of the red line close to the vertical black line.

  – Added subordinate sentence: ... measurement location which is highlighted by the black vertical line ...

- *p. 14-15, Figs. 7 and 8: Why are contour plots of Fig. 7 not as smooth as respective plots of Fig. 8. The caption mentions "filtered": aren't these "unfiltered"?*

  – As mentioned in the text: these contour plots are filtered by LiDAR availability. As a results, WRF data is discarded at times when LiDAR is not available. Therefore, the WRF results are skewed, but similar to LiDAR measurements.

- *p. 20, l. 3: You write "We chose": how do you control this?*

  – This has been addressed together with comments from the second Referee.

- *p. 21, l. 8: "Misalignment" is between TETHER and wind direction. It should be clear that the azimuth and elevation angles describe the angular position of the kite or aircraft with respect to the ground station. Renaming of $\Theta$ as elevation angle is dangerous, because it is generally use for the polar angle.*

  – Changed elevation angle to $\varepsilon$

  – Changed sentence to: Losses associated with mispositioning of the aircraft relative to the wind direction, expressed by azimuth angle $\phi$, elevation angle $\varepsilon$ relative to the ground station, are included in the model.

**0.2.6 Conclusions**

- *I am missing some conclusions of Section 4.7 on the AWES power estimation.*

  – reworked conclusion paragraph on AWES.

**0.3 Language and style comments**

**0.3.1 General spelling**

- *Use of dashes should be checked (e.g. "high-resolution data" or "long-term statistics" would be correct)*

  – hyphen added to the best of my knowledge

- *Do not capitalize abbreviations*
  *(see https:// www.aje.com/ en/ arc/ editing-tipcapitalization-when-defining-abbreviations/ ).*

  – removed capitalization in abbreviations except for names.

**0.3.2 Title**

- *p.1: "Airborne Wind Energy" should be "Airborne wind energy".*

  – removed capitalization, changed to "airborne wind energy"

**0.3.3 Abstract**

- *p. 1, l. 9: I would add an "it" between "but" and "becomes"*

  – pronoun "it" is omitted since there is no ambiguity that the topic of the sentence is the impact of nudging.

**0.3.4 Introduction**

- *p. 2, l. 4: "Airborne Wind Energy Systems" should be "Airborne wind energy systems".*

  – fixed capitalization

- *p. 2, l. 4: I would say that AWES are a class of renewable energy technologies, and not a source of energy. The source is the wind.*

  – Implemented.

- *p. 2, l. 10: Instead of "marketplace" I would just write "market".*

  – Implemented.

- *p. 2, l. 11: ...none are YET commercially available.*

  – changed to: ...none are currently commercially available

- *p. 2, l. 13: "power" should be "power output" as you list wind energy technologies here.*

  – Implemented.

- *p. 2, l. 23: ... variations to resolved quantities are parametrized. The "resolved" sounds wrong and the meaning of this sentence is also not clear to me.*

  – Corrected the sentence: Sub-gridscale high frequency variations **of** resolved quantities are parameterized.

- *p. 2, l. 25: "here presented" -> "presented in this study"*

  – Sentence removed in process of editing the manuscript.

- *p. 2, l. 27: Year is missing in reference.*

  – Reference updated

- *p. 3, l. 1: "power" -> "power output".*

  – Implemented.

0.3.5   Measurement campaign

- *p. 4, l. 2: Year is missing in reference.*

  – Reference updated

- *p. 4, l. 3: Should be "emphasizes".*

  – Implemented.

- *p. 4, l. 6: Should be "... the WRF-calculated...". The entire expression "WRFcal-culated sensible surface heat flux (SHF)" sounds incomprehensible to me. What is the role of the "sensible"?*

  – Added hyphen.
  – Latent and sensible heat are types of energy released or absorbed in the atmosphere.
    * "In meteorology, latent heat flux is the flux of heat from the Earth's surface to the atmosphere that is associated with evaporation or transpiration of water at the surface and subsequent condensation of water vapor in the troposphere." https://en.wikipedia.org/wiki/Latent_heat
    * "In meteorology, the term 'sensible heat flux' means the conductive heat flux from the Earth's surface to the atmosphere." https://en.wikipedia.org/wiki/Sensible_heat

- *p. 4, l. 9: Should be "... the SHF".*

  – Implemented.

- *I would move Fig. 1 to the next section and remove the reference to the white X here. Because in the next section you explain the 3 hierarchically nested domains used for the WRF. Here, in this section, the figure introduces more questions than answers.*

  – Figure was moved to section 2.
  – Reference to white X was kept to show the measurement location and the location where observations were implementation.

0.3.6  Mesoscale modeling Framework

- *p. 4, l. 14: Year is missing in reference. It is also not clear whether the "section 2" in the referenced paper or the present one is meant.*

- – Citation updated and section 2 removed.

- *p. 4, l. 15: Year is missing in reference.*

  - – Citation updated.

- *p. 4, l. 17: Why discussing here spatial resolutions when this is all given in Table 1?*

  - – Spatial resolution is kept in the sentence and table removed.

- *p. 4, l. 23: "Turbulent Kinetic Energy" should be "Turbulent kinetic energy". Add "(TKE)" here and use the abbreviation in the next sentence.*

  - – Replaced by "turbulent kinetic energy".

- *p. 4, l. 29: Maybe a footnote link with the URL is better? This bibliographic reference looks strange.*

  - – Removed citation and replaced with footnote: "EDDY: HPC cluster at the Carl von Ossietzky Universität Oldenburg, see: https://www.uni-oldenburg.de/fk5/wr/ hochleistungsrechnen/hpc-facilities/eddy/"

- *p. 5, l. 20: Something is wrong after $W_{xy}$.*

  - – Fixed equation variable

**0.3.7 Results**

- *p. 6, l. 5: Should be "differences".*

  - – Implemented.

- *p. 7, Fig. 2, legends: text and number should be separated by a space and also a comma.*

  – Updated legend to: linear regression, slope: 0.985.

- *p. 9, l. 2: Replace "bias" by "error".*

  – Implemented.

- *p. 17, l. 3: Reference missing (?).*

  – Citation updated.

- *p. 19, Table 2: last three columns in % of time would be better readable.*

  – Implemented.

- *p. 20, l. 4: "additional two" -> "two additional"*

  – Paragraph updated while editing.

- *p. 20, l. 5: Figure reference is missing (??).*

  – Paragraph updated while editing.

- *p. 21, l. 6: Why do you use a subscript "air" for the density? This study is only about atmospheric flows, so the index can be safely omitted.*

  – That is correct. However, we chose to keep the subscript "air" for clarity.

- *p. 21, l. 7: Set equation in displaymode.*

  – kept equation in text. Could be changed when typesetting the manuscript.

- *p. 21, l. 10: "... are assumed constant are ...": something is wrong here*

– Changed sentence to: " ... and drag coefficient ($c_D$ =0.06), which are assumed to be constant, are geometrically related to ..."

**0.3.8   Conclusions**

• *p. 23, l. 7: Six months OF LiDAR .....*

    – Implemented.

• *p. 23, l. 13: Dot behind "decreases" is missing.*

    – Paragraph changed while editing the manuscript.

**0.3.9   Appendices**

• *p. 25, Figure A1: what means the question mark at the end of this caption?  "U profile" -> "velocity profile".*

    – Removed figure.

• *p. 25-26, Table A1: Include in the caption to which software & version, possibly also model, these settings refer.*

    – Caption updated to: "Namelist parameters for WRF 3.6.1 observation nudging"

**0.3.10   References**

*There are many references for which the DOI is occurring twice, as "doi:..." and as URL "https://doi.org/...".*

removed URL and kept DOI where applicable.

- *p. 28, l. 13: what is this oCLC number? I would use either ISBN or DOI.*

  – removed oCLC and replaced with DOI.

- *p. 28, l. 19: Publisher or standard-issuing organization is missing.*

  – updated reference to ISO 2533:1975

- *p. 29, l. 15: Correct the URL.*

  – Citation replaced with footnote.

- *p. 29, l. 23: Add DOI.*

  – Citation correct, long, long list of contributers followed by a DOI and URL

- *p. 29, l. 27: Insert comma/dot and space between "2016" and "At".*

  – Not sure where. No 2016 in this citation.

- *p. 30, l. 3: "Statistik" should be starting with a capital "S", according to German spelling. Consider choosing an English textbook as reference.*

  – Replaced reference.

- *p. 30, l. 8: This is a contributed chapter in a book. Accordingly the reference should be Schmehl, R., Noom, M., van der Vlugt, R.: Traction Power Generation with Tethered Wings. In: Ahrens, U., Diehl, M., Schmehl, R. (eds.) Airborne Wind Energy. Springer, Berlin Heidelberg, 2013.*

  – The citation is taken directly from: https://link.springer.com/chapter/10.1007/978-3-642-39965-7_2 . Had to change @inbook to @incollection to make it work.

- *p. 30, l. 14: Update this*

– Updated.

Please also note the supplement to this comment:
https://www.wind-energ-sci-discuss.net/wes-2019-7/wes-2019-7-AC2-supplement.pdf

**Supplement:**

**Improving mid-altitude mesoscale wind speed forecasts using LiDAR-based observation nudging for airborne wind energy systems**

Markus Sommerfeld1, Curran Crawford1, Gerald Steinfeld2, and Martin Dörenkämper3

1Institute for Integrated Energy Systems, University of Victoria,British Columbia, Canada 2Institute of Physics-Energy Meteorology, Carl von Ossietzky Universität Oldenburg, Germany 3Fraunhofer Institute for Wind Energy Systems, Oldenburg, Germany

Correspondence to: Markus Sommerfeld (msommerf@uvic.ca)

**Abstract.**

[revised manuscript text omitted]

Unstable and stable stratifications were identified by partitioning the data based on the sign of WRF-calculated sensible surface heat flux (SHF). These two atmospheric conditions lead to bi-modal wind speed probability distributions aloft which is

30 not adequately represented by a single Weibull distribution fit. Mid-altitude wind speeds are better represented by the weighted sum of two wind speed probability density function (PDF) fits conditioned by the sign of the SHF.

Figure 1. Topography map of all three WRF model domains (a) and a magnification of the innermost domain (b) with the LiDAR measurement site marked by a white X.

**3 Mesoscale Modeling Framework**

To complement the 6 months LiDAR data set two WRF (v. 3.6.1) simulations using the advanced research weather research and forecasting (ARW) model (Skamarock and Klemp, 2008) were carried out. The 'baseline run' (, which is hereinafter referred to as *NoOBS*, ) is a 12 month study of the area around the measurement location (see figure 2) from the 1st of September 2015 used to derive annual statistics. LiDAR measurements (Sommerfeld et al., 2019) (see section 2) were incorporated into the six months test model between September 2015 and February 2016 using *OBSGRID* (Wang et al., 2015) (, which is hereinafter

5 months test model between September 2015 and February 2016 using *OBSGRID* (Wang e referred to as *OBS*).

This methodology uses the difference between model and measurements to calculate a non-physical forcing term that which is added to the governing conservation equations of the simulation to gradually nudges the model towards the observation (see equation 1) (Stauffer et al., 1991; Deng et al., 2007). Each simulation is composed of three nested domains with 27-, 9-

- 10 and 3-km grid spacing and horizontal grid dimensions of about  $120 \times 120$  elements at 60 pressure heights along the terrain following vertical hybrid pressure coordinate  $\eta$ . Differences between the simulation runs (see section 3.1) are compared within the innermost domain of the simulation. Output data was stored in 10 min intervals. Figure 2 shows the topography map of the simulation. Initial and boundary conditions of both simulations are based on the *ERA-Interim* (Dee et al., 2011) reanalysis data set by the European centre for medium-range weather forecasts (ECMWF) which consists of 6 hourly atmospheric fields with a
- 15 spatial resolution of roughly 80 km horizontally and 60  $\eta$  levels. Turbulent Kinetic Energy closure within the ABL was achieved

---

## Author Comment (AC3) · 29 Jun 2019

Dear Dr. Bechtle,

Thank you very much for your helpful comments to our manuscript, "Improving mesoscale wind speed forecasts using LiDAR-based observation nudging for Airborne Wind Energy Systems", wes-2019-7. We have clarified the definition of RMSE and added additional reference to technical document describing the LiDAR uncertainty. Following are our replies to your comments and a description of modification to the manuscript.

Changes are highlighted in the "Supplementary Material" pdf. Text and figures marked in red were removed from the original submission and replaced by text and figures marked in blue. Following are our replies to your comments and a description of modification to the manuscript.

Sincerely,
Markus Sommerfeld

**Comments by the Referee**

**0.1 General comments**

*Dear Authors, congratulations for your very interesting and important analysis. I read the paper with great interest. I have one or two questions about the methods which lead to Fig. 3.*

*First, I am unsure about what you mean by "The continuous line in the left sub-figure represents the Root Mean Square Error (RMSE) of wind speed." This is just a question for clarification... The RMS of what? is it a measure of the temporal variation of the measured or simulated wind speed on its own within some interval? Or an estimate of the precision of simulation or measurement (and if that, how is that uncertainty derived?)? Or an RMS of a difference between different quantities? I think the paper would profit if you could explain this in more detail.*

- A clearer definition of wind speed RMSE, which quantifies the error between LiDAR measurements and WRF simulations, has been added to the paper.

*The other question, which maybe is connected to the first question, is about the uncertainty. Your measurement itself is subject to an uncertainty, and it would be interesting if that would be clearly described.*

*Thank you very much! Philip Bechtle*

- We have added a reference to a technical report verifying the performance of the used LiDAR (https://www.woodgroup.com/__data/assets/pdf_file/0023/15692/report_Sgurr_

20130529_FINAL1.pdf). Further information can be found in our previous paper "Li-DAR based characterization of mid altitude wind conditions for airborne wind energy systems" (https://doi.org/10.1002/we.2343).

Please also note the supplement to this comment:
https://www.wind-energ-sci-discuss.net/wes-2019-7/wes-2019-7-AC3-supplement.pdf

---

## Author Comment (AC4) · 29 Jun 2019

Dear Dr. Saraceno,

Thank you very much for your comments on our manuscript, "Improving mesoscale wind speed forecasts using LiDAR-based observation nudging for Airborne Wind Energy Systems", wes-2019-7.

Our analyses focus on wind conditions up to 1100 m and apply a simplified ground-gen AWES model to estimate maximum power output and optimal altitude for this maximum power (neglecting tether weight and drag). These findings are representative for a specific location and time period in Northern-Germany, based on mesoscale wind and weather simulations and backed by a published study on LiDAR measurements at this specific location("LiDAR based characterization of mid altitude wind conditions for airborne wind energy systems" (https://doi.org/10.1002/we.2343).

Comparing different AWES designs and concepts is beyond the scope of this paper. We do not state that designing a system for maximum power and neglecting capacity factor is desirable. On the contrary, we hope that our findings, which are to our knowledge the best representation of actual wind conditions at higher altitudes, will educate designers, manufacturers and researchers to design more efficient and reliable systems.

Sincerely,
Markus Sommerfeld